# FORMAL INTERPRETABILITY WITH MERLIN-ARTHUR CLASSIFIERS

## ABSTRACT

We propose a new type of multi-agent interactive classifier that provides provable interpretability guarantees even for complex agents such as neural networks. These guarantees consist of bounds on the mutual information achieved by the features selected by this classifier. Our results are inspired by the Merlin-Arthur protocol from Interactive Proof Systems and express this bound in terms of measurable metrics such as soundness and completeness. We introduce the notion of Asymmetric Feature Concentration which relates the information carried by sets of features to the one of individual features. Crucially, our bound does not rely on optimal play by the agents nor on independently distributed features. We verify our framework through numerical experiments on image classification problems.

## 1 INTRODUCTION

Safe deployment of Neural Network (NN) based AI systems in high-stakes applications requires that their reasoning be subject to human scrutiny. The field of Explainable AI (XAI) has thus put forth a number of interpretability approaches, among them saliency maps (Mohseni et al., 2021) and mechanistic interpretability (Olah et al., 2018). These have had some successes, such as detecting biases in established datasets (Lapuschkin et al., 2019), or connecting individual neurons to understandable features (Carter et al., 2019). However, these approaches are motivated purely by heuristics and come without any theoretical guarantees. Thus, their success cannot be verified. It has also been demonstrated for numerous XAI-methods that they can be manipulated by a clever design of the NNs (Slack et al., 2021; 2020; Anders et al., 2020; Dimanov et al., 2020). On the other hand, formal approaches to interpretability run into complexity barriers when

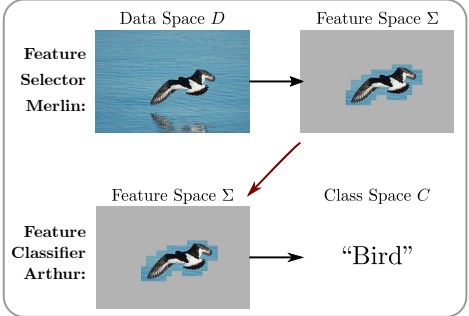

Merlin-Arthur Classifier

Figure 1: The Merlin-Arthur classifier consists of two interactive agents that communicate over an exchanged feature. This feature serves as an interpretation of the classification.

applied to NNs and require an exponential amount of time to guarantee useful properties (Macdonald et al., 2020; Ignatiev et al., 2019). This makes any "right to explanation," as codified in the EU's *GDPR* (Goodman & Flaxman, 2017), unenforceable.

In this work, we design a classifier that guarantees feature-based interpretability under reasonable assumptions, thus overcoming both theoretical and computational shortcomings. For this, we connect classification to the Merlin-Arthur protocol (Arora & Barak, 2009) from Interactive Proof Systems (IPS), see Figure 1. For easier illustration, we split the unreliable prover into a cooperative Merlin and an adversarial Morgana. Merlin aims to send features that cause Arthur to correctly classify the underlying data point. On the opposite side, Morgana selects features to convince Arthur of the wrong class. Arthur does not know who sent the feature and is allowed to say "Don't know!" if he cannot discern the class. We can then translate the concepts of *completeness* and *soundness* from IPS to our setting. Completeness describes the probability that Arthur classifies correctly based on features from Merlin. Soundness is the probability that Arthur does not get fooled by Morgana, thus

either giving the correct class or answering "Don't know!". These two quantities can be measured on a test dataset and are used to lower bound the information contained in features selected by Merlin.

## 1.1 Related Work

Interactive classification has emerged as part of the effort to design inherently interpretable classifiers Lei et al. (2016); Li et al. (2016); Chen et al. (2018); Bastings et al. (2019). In this setup, the feature selector chooses a feature from a data point and presents it to the classifier who decides the class, see Figure 1. The classification accuracy is meant to guarantee the informativeness of the exchanged features.

However, it was noted by Yu et al. that the selector and the classifier can cooperate to achieve high accuracy while communicating over uninformative features, see Figure 2 for an illustration of this "cheating". Thus, one cannot hope to bound the information content of features via accuracy alone. The authors propose to include an adversarial feature classifier to remedy this fact, however do not provide any bounds. Irving et al. introduce a different adversary, and we discuss in Appendix A.4 why this approach cannot yield bounds similar to the ones in our work. Chang et al. include an adversarial selector to prevent the cheating. The reasoning is that any "cheating" strategy can be exploited by the adversary to fool the classifier into stating the wrong

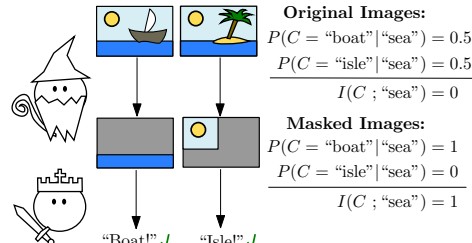

Figure 2: Illustration of "cheating" behaviour. In the original dataset, the features "sea" and "sky" appear equally in both classes "boat" and "island". In the new set of images that Merlin creates by masking features of the original image, the "sea" feature is visible only in the images labelled "boat" and the "sky" feature is visible only in the images labelled "island". Thus, these features now strongly indicate the class of the image. This allows Merlin to communicate the correct class with uninformative features — in contrast to our concept of an interpretable classifier.

class, see Figure 3 for an illustration. Anil et al. investigate scenarios in which this three-player setup converges to an equilibrium of perfect completeness and soundness. However, both works assume that the players can play optimally. This, however, is unlikely in practice, since the general feature selection problem is hard Waeldchen et al. (2021). Optimal play would thus amount to an exponentially expensive search problem similar to Chattopadhyay et al. (2022). In our work, we rather rely on the relative strength of the cooperative and the adversarial provers which can both be realised with imperfect strategies.

It has been shown that the optimal strategy for the prover is to select the features that have the highest mutual information with the class Chang et al. (2019). This demonstrates a strong theoretical link between a game equilibrium and the feature quality. However, the authors rely on the assumption that the features are independently distributed in each class, but in almost all types of data there are strong correlations between features. We do not make this assumption, but rather define Asymmetric Feature Correlation as an important property of datasets to draw conclusions about feature quality.

## 1.2 Contribution

Our framework requires few constraints on the classifier and is thus applicable to a wide range of tasks. Similarly, our results hold for general feature spaces, allowing free choice for the practitioner, e.g., parts of an image up to a certain size, or functional queries about the input.

1. We prove a lower bound for the mutual information of the exchanged features with the true class of the data point. This bound does not rely on the features being independently distributed, nor on restricting the strategy of the provers.

2. We introduce Asymmetric Feature Concentration as a possible effect that complicates drawing conclusions about individual features using the informativeness of feature sets. We show how to circumvent it in Theorem 2.8 or include it explicitly as in Theorem 2.10.

3. We numerically demonstrate how the interactive setup prevents the manipulation that has been demonstrated for other XAI-methods. Furthermore, we evaluate our theoretical bounds on the MNIST dataset for provers based on either Frank-Wolfe optimisers or UNets.

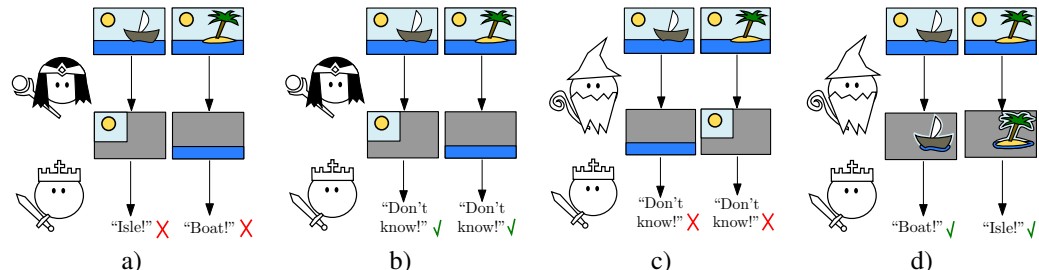

Figure 3: Strategy evolution with Morgana. a) Due to the "cheating" strategy from Figure 2, Arthur expects the "sea" feature for boats and the "sky" for islands. Morgana can exploit the situation and send the "sky" feature to trick Arthur into classifying a "boat" image as an "island" (and vice versa with the "sea" feature). b) To not be fooled into stating the wrong class when represented with an ambiguous feature, Arthur refrains from giving a concrete classification. c) Since Arthur does not know who sends the features, he now cannot leverage the uninformative features sent by Merlin. d) Merlin adapts his strategy to only send unambiguous features that cannot be used by Morgana to fool Arthur.

## 2  THEORETICAL FRAMEWORK

What reasonably constitutes a feature strongly depends on the context. In Interpretability research, one often considers cutouts from the original data point, but also more general functional queries. Our results do not depend on the specific form of the feature. We thus opt to leave the definition abstract and define a feature as the set of data points that contain it. This also has the advantage to let us reason about feature spaces in terms of graphs. We make use of this in Appendix A.4 and proof via counterexample that the debate setup of Irving et al. (2018), where Arthur has access to features of both Merlin and Morgana, cannot lead to results of the same generality as ours.

**Definition 2.1** (Feature Space). *Given a dataset $D$, we call a set $\Sigma \subseteq 2^D$ a* feature space *on $D$ if (i) all data points have at least one feature, i.e. $\cup \Sigma = D$, (ii) the empty set is contained as a default feature, i.e. $\varnothing \in \Sigma$, (iii) there are only finitely many features per data point i.e., $\max_{\mathbf{x} \in D} |\{\phi \in \Sigma \,|\, \mathbf{x} \in \phi\}| < \infty$.*

The set $D$ is possibly infinite, e.g. the set of all images of hand-written digits. $\mathcal{D}$ is a distribution on this set. The finite training and test sets, e.g. MNIST, for our algorithms are assumed to be faithful samples from this distribution. We say a data point $\mathbf{x}$ *contains* a feature $\phi \in \Sigma$ if $\mathbf{x} \in \phi$.

One important practical example of a feature space is given in Example 2.2. Every feature corresponds to a subset of pixels $S$ of size $k$ in an image $\mathbf{x}$, and any data point $\mathbf{y}$ that has the same values on the subset, i.e., $\mathbf{y}_S = \mathbf{x}_S$, contains the same feature.

**Example 2.2** (Features as partial vectors of size $k$). *Let $D \subset [0,1]^d$. For $k \in [d]$, we define a feature space corresponding to partial vectors with support of size $k$ subsets as*

$$\Sigma = \bigcup_{\mathbf{x} \in D} \bigcup_{S \subset [d], |S| = k} \{\{\mathbf{y} \in D \,|\, \mathbf{y}_S = \mathbf{x}_S\}\} \cup \{\varnothing\}.$$

We now define a data space. For notational simplicity, we restrict ourselves to two classes.

**Definition 2.3** (Two-class Data Space). *We consider the tuple $\mathfrak{D} = ((D, \sigma, \mathcal{D}), c, \Sigma)$ a* two-class data space *consisting of the dataset $D \subseteq [0,1]^d$, sigma algebra $\sigma$ and probability distribution $\mathcal{D}$ along with the ground truth class map $c : D \to \{-1, 1\}$ and the feature space $\Sigma$.*

*The* class imbalance *$B$ of a two-class data space is $\max_{l \in \{-1,1\}} \mathbb{P}_{\mathbf{x} \sim \mathcal{D}}[c(\mathbf{x}) = l] / \mathbb{P}_{\mathbf{x} \sim \mathcal{D}}[c(\mathbf{x}) = -l]$.*

Note that our assumption on the ground truth class map requires that there be a unique output for any given input. This may not necessarily be true for many data sets. We now introduce the notion of a feature selector (as prover) and feature classifier (as verifier).

**Definition 2.4** (Feature Selector). *For a given dataset $D$, we call $M$ a* feature selector, *if*

$$M : D \to \Sigma \quad s.t. \quad \forall \mathbf{x} \in D : (\mathbf{x} \in M(\mathbf{x})) \vee (M(\mathbf{x}) = \varnothing),$$

*which means that for every data point $\mathbf{x} \in D$ the feature selector $M$ chooses a feature that is present in $\mathbf{x}$ or returns the empty set. We call $\mathcal{M}(D)$ the space of all feature selectors for a dataset $D$.*

**Definition 2.5** (Feature Classifier). *We define a* feature classifier *for a feature space $\Sigma$ as a function $A : \Sigma \to \{-1, 0, 1\}$. Here, 0 corresponds to the situation where the classifier is unable to identify a correct class. We call the space of all feature classifiers $\mathcal{A}$.*

## 2.1 INFORMATION MEASURES: CONDITIONAL ENTROPY AND AVERAGE PRECISION

We consider a feature to carry class information if it has high mutual information with the class. For a given feature $\phi \in \Sigma$ and a data point $\mathbf{y} \sim \mathcal{D}$ the mutual information is defined as

$$I_{\mathbf{y} \sim \mathcal{D}}(c(\mathbf{y}); \mathbf{y} \in \phi) := H_{\mathbf{y} \sim \mathcal{D}}(c(\mathbf{y})) - H_{\mathbf{y} \sim \mathcal{D}}(c(\mathbf{y}) \,|\, \mathbf{y} \in \phi).$$

When the conditional entropy $H_{\mathbf{y} \sim \mathcal{D}}(c(\mathbf{y}) \,|\, \mathbf{y} \in \phi)$ goes to zero, the mutual information becomes maximal and reaches the pure class entropy $H_{\mathbf{y} \sim \mathcal{D}}(c(\mathbf{y}))$ which measures how uncertain we are about the class a priori. A closely related concept is *precision*. Given a data point $\mathbf{x}$ with feature $\phi$, precision is defined as $\mathbb{P}_{\mathbf{y} \sim \mathcal{D}}[c(\mathbf{y}) = c(\mathbf{x}) \,|\, \mathbf{y} \in \phi]$ and was introduced in the context of interpretability by Ribeiro et al. (2018) and Narodytska et al. (2019). It can be used to bound conditional entropy and thus mutual information. We now extend this definition to a feature selector.

**Definition 2.6** (Average Precision). *For a given two-class data space $((D, \sigma, \mathcal{D}), c, \Sigma)$ and a feature selector $M \in \mathcal{M}(D)$, we define the* average precision *of $M$ with respect to $\mathcal{D}$ as*

$$Q_{\mathcal{D}}(M) := \mathbb{E}_{\mathbf{x} \sim \mathcal{D}}[\mathbb{P}_{\mathbf{y} \sim \mathcal{D}}[c(\mathbf{y}) = c(\mathbf{x}) \,|\, \mathbf{y} \in M(\mathbf{x})]].$$

The average precision $Q_{\mathcal{D}}(M)$ can be used to bound the *average* conditional entropy, defined as

$$H_{\mathbf{x}, \mathbf{y} \sim \mathcal{D}}(c(\mathbf{y}) \,|\, \mathbf{y} \in M(\mathbf{x})) := \mathbb{E}_{\mathbf{x} \sim \mathcal{D}}[H_{\mathbf{y} \sim \mathcal{D}}(c(\mathbf{y}) \,|\, \mathbf{y} \in M(\mathbf{x}))],$$

and accordingly the average mutual information. For greater detail, see Appendix B. We can connect $Q_{\mathcal{D}}(M)$ to the precision of any feature selected by $M$ in the following way.

**Lemma 2.7.** *Given $\mathfrak{D} = ((D, \sigma, \mathcal{D}), c, \Sigma)$ , a feature selector $M \in \mathcal{M}(D)$ and $\delta \in [0, 1]$. Let $\mathbf{x} \sim \mathcal{D}$, then with probability $1 - \delta^{-1}(1 - Q_{\mathcal{D}}(M))$, $M(\mathbf{x})$ selects a feature $\phi$, s.t.*

$$\mathbb{P}_{\mathbf{y} \sim \mathcal{D}}[c(\mathbf{y}) = c(\mathbf{x}) \,|\, \mathbf{y} \in \phi] \geq 1 - \delta.$$

The proof follows directly from Markov's inequality, see Appendix B. We will now introduce a new framework that will allow us to prove bounds on $Q_{\mathcal{D}}(M)$ and thus assure feature quality. For $I$ and $H$, we will leave the dependence on the distribution implicit when it is clear from context.

## 2.2 MERLIN-ARTHUR CLASSIFICATION

For a feature classifier $A$ (Arthur) and two feature selectors $M$ (Merlin) and $\widehat{M}$ (Morgana) we define

$$E_{M, \widehat{M}, A} := \left\{ x \in D \,\Big|\, A(M(\mathbf{x})) \neq c(\mathbf{x}) \vee A\left(\widehat{M}(\mathbf{x})\right) = -c(\mathbf{x}) \right\}, \tag{1}$$

as the set of data points for which Merlin fails to convince Arthur of the correct class or Morgana is able to trick him into giving the wrong class, in short, the set of points where Arthur fails. We can now state the following theorem connecting the competitive game between Arthur, Merlin and Morgana to the class conditional entropy.

**Theorem 2.8.** *[Min-Max] Let $M \in \mathcal{M}(D)$ be a feature selector and let*

$$\epsilon_M = \min_{A \in \mathcal{A}} \max_{\widehat{M} \in \mathcal{M}} \mathbb{P}_{\mathbf{x} \sim \mathcal{D}}\Big[\mathbf{x} \in E_{M, \widehat{M}, A}\Big].$$

*Then there exists a set $D' \subset D$ with $\mathbb{P}_{\mathbf{x} \sim \mathcal{D}}[\mathbf{x} \in D'] \geq 1 - \epsilon_M$ such that for $\mathcal{D}' = \mathcal{D}|_{D'}$ we have*

$$Q_{\mathcal{D}'}(M) = 1 \quad \text{and thus} \quad H_{\mathbf{x}, \mathbf{y} \sim \mathcal{D}'}(c(\mathbf{y}) \,|\, \mathbf{y} \in M(\mathbf{x})) = 0.$$

The proof is in Appendix B. This theorem states that if Merlin's strategy allows Arthur to classify almost perfectly, i.e., small $\epsilon_M$, then there exists a dataset that covers almost the entire original dataset and on which the class entropy conditioned on the selected features is zero. Note that these guarantees are for the set $D'$ and not the original set $D$. A bound for the set $D$, such as $Q_{\mathcal{D}}(M) \geq 1 - \epsilon_M$, is complicated by a principle we call *asymmetric feature concentration (AFC)*.

**Definition 2.9** (Asymmetric feature concentration). *Let $((D, \sigma, \mathcal{D}), c, \Sigma)$ be a two-class data space, then the asymmetric feature concentration $\kappa$ is defined as*

$$\kappa = \max_{l \in \{-1, 1\}} \max_{F \subset \Sigma} \mathbb{E}_{\mathbf{y} \sim \mathcal{D}_l|_{\cup F}} \left[ \max_{\substack{\phi \in F \\ s.t.\ \mathbf{y} \in \phi}} \frac{\mathbb{P}_{\mathbf{x} \sim \mathcal{D}_{-l}}[\mathbf{x} \in \phi \,|\, \mathbf{x} \in \cup F]}{\mathbb{P}_{\mathbf{x} \sim \mathcal{D}_l}[\mathbf{x} \in \phi \,|\, \mathbf{x} \in \cup F]} \right].$$

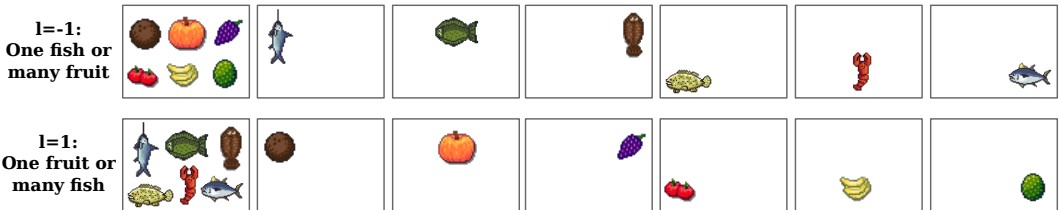

Figure 4: Example of a dataset an AFC $\kappa = 6$. The "fruit" features are concentrated in one image for class $l = -1$ but spread out over six images for $l = 1$ (vice versa for the "fish" features). Each individual feature is not indicative of the class as it appears exactly once in each class. Nevertheless, Arthur and Merlin can exchange "fruits" to indicate "$l = 1$" and "fish" for "$l = -1$". The images where this strategy fails or can be exploited by Morgana are the two images on the left. Applying Theorem 2.8, we get $\epsilon_M = \frac{1}{7}$ and the set $D'$ corresponds to all images with a single feature. Restricted to $D'$, the features determine the class completely.

**Asymmetric Feature Correlation:** AFC describes a possible quirk of datasets, where a set of features is strongly concentrated in a few data points in one class and spread out over almost all data points in another. We give an illustrative example in Figure 4, and derive the formal definition in Appendix B. If a data space $\mathfrak{D}$ has a large AFC $\kappa$, Merlin can use features that individually appear equally in both classes (low precision) to indicate the class where they are spread over almost all points. Morgana can only fool Arthur in the other class where these features are highly concentrated, thus only in a few data points. This ensures a small $\epsilon_M$ even with uninformative features.

Though it is difficult to calculate the AFC for complex datasets, we show that it can be bounded above by the maximum number of features per data point in $D$.

**Lemma 2.10.** *Let $\mathfrak{D} = ((D, \sigma, \mathcal{D}), c, \Sigma)$ be a two-class data space with AFC of $\kappa$. Let $K = \max_{\mathbf{x} \in D} |\{\phi \in \Sigma \,|\, \mathbf{x} \in \phi\}|$ be the maximum number of features per data point. Then $\kappa \leq K$.*

We explain further in Appendix B. The number $K$ depends on the type of features one considers.

## 2.3 REALISTIC ALGORITHMS AND RELATIVE SUCCESS RATE

In Theorem 2.8, we assume a perfect Morgana, which is only practical for simple datasets. Consider the UCI Census Income dataset Dua & Graff (2017) with 14 dimensions. When restricting features to a maximal size of seven, the search space is at most $\binom{14}{7} = 3432$, within range for exhaustive search. Contrary to this, modelling the UCI data distribution is still an involved task, and when done incorrectly, leads to incorrect explanations Frye et al. (2020). Our approach is therefore advantageous for low-dimensional datasets often used in recruitment and criminal justice, where interpretability is crucial. However, we also aim to apply our setup to high-dimensional datasets, where exhaustive search is not possible.

The feature selection problem is as follows. Given $\mathbf{x} \in D$ and a feature classifier $A$, we define

$$M^*(\mathbf{x}, l) := \arg\max_{\phi \in \Sigma} \, l \cdot A(\phi). \tag{2}$$

Here, $M^*$ returns a mask that convinces Arthur of class $l$ if possible. Given a fixed Arthur, $M(\mathbf{x}) = M^*(\mathbf{x}, c(\mathbf{x}))$ and $\widehat{M}(\mathbf{x}) = M^*(\mathbf{x}, -c(\mathbf{x}))$ represent the optimal strategy for Merlin and Morgana, respectively. Henceforth, we will consider algorithms that allow Merlin and Morgana to approximate $M^*$. Their relative strength is captured in the following notion:

**Relative Success Rate:** Morgana only has to find features successfully found by Merlin, albeit in a different class. The question is how much the rest of the features, which are not indicative of the class, influence the probability of success. In Appendix B, we derive the notion of *relative success rate $\alpha$*, i.e., the success rate of Morgana relative to Merlin for finding the same features. We give an artificial example of a worst-case data space that illustrates this idea in Figure 13. We argue that realistically, we can ensure a large $\alpha$ by using an algorithm for Morgana that is at least as powerful as the one for Merlin. Together with the AFC, this allows us to state the following theorem.

**Definition 2.11** (Relative Success Rate). *Let $\mathfrak{D} = ((D, \sigma, \mathcal{D}), c, \Sigma)$ be a two-class data space. Let $A \in \mathcal{A}$ and $M : D \times \{-1, 1\} \to \Sigma^d$ an approximation for $M, \widehat{M} \in \mathcal{M}(D)$. Then the relative success rate $\alpha$ of $\widehat{M}$ with respect to $A, M$ and $\mathfrak{D}$ is defined as*

$$\alpha := \min_{l \in \{-1, 1\}} \min_{F \subseteq A^{-1}(l)} \frac{\mathbb{P}_{\mathbf{x} \sim \mathcal{D}_{-l}}\left[A(\widehat{M}(\mathbf{x})) = l \mid \mathbf{x} \in \cup F\right]}{\mathbb{P}_{\mathbf{x} \sim \mathcal{D}_l}[M(\mathbf{x}) \in F \mid \mathbf{x} \in \cup F]}.$$

**Theorem 2.12.** *Let $\mathfrak{D} = ((D, \sigma, \mathcal{D}), c, \Sigma)$ be a two-class data space with AFC of $\kappa$ and class imbalance $B$. Let $A \in \mathcal{A}$, and $M, \widehat{M} \in \mathcal{M}(D)$ such that $\widehat{M}$ has a relative success rate of $\alpha$ with respect to $A, M$ and $\mathfrak{D}$. Define*

*1. Completeness:*    $\min\limits_{l \in \{-1, 1\}} \mathbb{P}_{\mathbf{x} \sim \mathcal{D}_l}[A(M(\mathbf{x})) = c(\mathbf{x})] \geq 1 - \epsilon_c$,

*2. Soundness:*    $\max\limits_{l \in \{-1, 1\}} \mathbb{P}_{\mathbf{x} \sim \mathcal{D}_l}\left[A\left(\widehat{M}(\mathbf{x})\right) = -c(\mathbf{x})\right] \leq \epsilon_s$,

*where $\mathcal{D}_l$ is the data distribution conditioned on the class $l$. Then it follows that*

$$Q_{\mathcal{D}}(M) \geq 1 - \epsilon_c - \frac{\kappa \alpha^{-1} \epsilon_s}{1 - \epsilon_c + \kappa \alpha^{-1} B^{-1} \epsilon_s}.$$

The proof is provided in Appendix B. The core assumption we make when comparing our lower bound with the measured average precision in Section 3 is the following. For our numerical bounds to make sense we have to rely on the following assumption:

**Assumption 2.13.** *The AFC $\kappa$ of $\mathfrak{D}$ and the relative success rate $\alpha$ of $\widehat{M}$ w.r.t. $A$, $M$, $\mathfrak{D}$ are $\mathcal{O}(1)$.*

Both $\kappa$ and $\alpha$ are difficult to estimate for high-dimensional datasets, but they offer clear guidelines on how to think about your dataset and algorithms in terms of interpretability. We numerically evaluate this assumption in Section 3.2 on a dataset simple enough to actually calculate $Q_{\mathcal{D}}(M)$.

**Finitely Sampled and Biased Dataset**    We usually have access to only finitely many samples of a dataset. Additionally, the observed samples can be biased as compared to the true distribution. We prove bounds for both cases in Appendix B.5. We show that any exchanged feature is either informative, or it is incorrectly represented in the dataset—thus highlighting the bias!

## 3    NUMERICAL IMPLEMENTATION

We illustrate the merit of our framework with experiments on the UCI Census Income dataset and on the MNIST handwritten digits dataset (LeCun et al., 1998).

First, we describe how to train the agents Arthur, Merlin, and Morgana in a general $n$-class interactive learning setting for image data of dimension $d$, where $n, d \in \mathbb{N}$. We explain in Appendix A.5 why we chose a multi-class neural network for Arthur and compare with the approaches of Chang et al. (2019) and Anil et al. (2021). The training process for tabular data, like the Census dataset, is equivalent and a detailed overview is provided in the appendix.

Arthur is modelled by a feed-forward neural network. He returns a probability distribution over his possible answers, so let $A : [0, 1]^d \to [0, 1]^{(n+1)}$, corresponding to the probabilities of stating a class or "I don't know". The provers select a set $S$ of at most $k$ pixels from the image via a *mask* $\mathbf{s} \in B_k^d$, where $B_k^d$ is the space of $k$-sparse binary vectors of dimension $d$. A masked image $\mathbf{s} \cdot \mathbf{x}$ has all its pixels outside of $S$ set to a baseline or a random value. We define the Merlin-loss $L_M$ as the cross-entropy loss with regard to the correct class, whereas the Morgana-loss $L_{\widehat{M}}$ considers the total probability of either answering the correct class or the "I don't know" option, so

$$L_M(A, \mathbf{x}, \mathbf{s}) = -\log\big(A(\mathbf{s} \cdot \mathbf{x})_{c(\mathbf{x})}\big) \quad \text{and} \quad L_{\widehat{M}}(A, \mathbf{x}, \mathbf{s}) = -\log\big(A(\mathbf{s} \cdot \mathbf{x}))_0 + A(\mathbf{s} \cdot \mathbf{x})_{c(\mathbf{x})}\big).$$

The total loss for Arthur is then

$$L_{M, \widehat{M}}(A) = \mathbb{E}_{\mathbf{x} \sim \mathcal{D}}\Big[(1 - \gamma)L_M(A, \mathbf{x}, M(\mathbf{x})) + \gamma L_{\widehat{M}}(A, \mathbf{x}, \widehat{M}(\mathbf{x}))\Big],$$

where $\gamma \in [0, 1]$ is a tunable parameter. In our experiments, we choose $\gamma > 0.5$ since we always want to ensure good soundness. We compare different $\gamma$ in Appendix C. Note that Merlin wants to minimise $L_M$, whereas Morgana aims to maximise $L_{\widehat{M}}$. In an ideal world, they would solve

$$M(\mathbf{x}) = \underset{\mathbf{s} \in B_k^d}{\arg\min}\, L_M(A, \mathbf{x}, \mathbf{s}) \quad \text{and} \quad \widehat{M}(\mathbf{x}) = \underset{\mathbf{s} \in B_k^d}{\arg\max}\, L_{\widehat{M}}(A, \mathbf{x}, \mathbf{s}). \tag{3}$$

The above solutions can be obtained either by solving the optimisation problem (Frank-Wolfe solver (Macdonald et al., 2021)) or by using NNs to predict the solutions (UNet architecture). We kept our training routine simple by looping over the three agents and describe the details of the training algorithm and architectures in Appendix C.

## 3.1 Preventing Manipulation

Since existing XAI algorithms come without guarantees, they cannot exclude the possibility for manipulation. Indeed, arbitrarily changing the interpretation by slightly changing the classifier has been demonstrated for numerous XAI approaches. Slack et al. fool LIME and ShAP by making use of the fact that these methods sample off-manifold evaluations of the classifier. We are robust against this approach, since the Merlin-Arthur classifier only takes on-manifold inputs. Dimanov et al., Heo et al. and Anders et al. optimise manipulated classifier networks to give the desired target explanations by penalizing any deviation. The equivalent for our setup is to put a penalty on Merlin to hide the true (potentially biased) explanations in the exchanged features. This scheme is unsuccessful for a Merlin-Arthur classifier. Indeed, for this setup we show numerically that either: (i) The bias becomes visible, (ii) Morgana can exploit the setup, i.e., soundness is low or (iii) the completeness, i.e., prediction accuracy, drops.

We demonstrate this on a modified UCI Census Income dataset[1]. We train a classifier to discriminate by "sex"—one of the remaining 12 features for every data point. We change the Merlin-loss to

$$L_M'(A, \mathbf{x}, \mathbf{s}) := L_M(A, \mathbf{x}, \mathbf{s}) + \beta\|\mathbf{s}_{\text{sex}}\|,$$

where $\beta > 0$, to train Merlin to hide the sensitive "sex" feature. We then train the Merlin-Arthur classifier to accept only men and set $k = 1$ to only send one feature. We compare a setup where soundness is ignored to one where we choose $\gamma$ such that a soundness of at least 90% is achieved. Note that the class map for the Census Income data set is not unique, i.e., there are duplicate outcomes for exactly the same input. However, only small fraction of data points (0.6%) have this issue.

The results are illustrated in Figure 5. When soundness is ignored ($\gamma = 0$), setting $\beta = 3.5$ causes Merlin to hide the "sex" feature and communicate "sex" via selecting "education" for men, "working hours" for women. When soundness is required ($\gamma = 0.67$), Merlin starts to reveal the "sex" feature again. Increasing the penalty to $\beta = 10$, causes him to hide "sex", but now the prediction accuracy reduces to $0.45$, i.e., close to chance and thus ceasing the discrimination. When high soundness is required, Merlin is unable to both discriminate and hide his bias.

## 3.2 Evaluation of Theoretical Bounds

We illustrate our framework on the MNIST dataset restricted to two classes and present the results as evaluated on the test dataset. We use Theorem 2.12 to calculate a lower bound on the average precision $Q_{\mathcal{D}}(M)$ for two different pairs of classes. We compare three different setups of feature selectors, one with Frank-Wolfe optimisers (**Opt**), one where Merlin is a UNet and Morgana an optimiser (**Hybrid**), and one where we use UNets for both (**Net**). For the purpose of numerical experiments, we assume that the MNIST dataset has an AFC of 1. This is needed for numerically calculating the lower bound on average precision. It implies that features which individually provide evidence for a particular class do not cumulatively provide evidence for a different class.

In Figure 6, we see that the lower bound is tight for larger masks, but drops off when $k$ is small. One reason for this is that for small masks, Arthur tends to give up on one of the classes, while keeping the completeness in the other class high. Since the bound considers the minimum over both classes,

---

[1]For this experiment, we have removed the "relationship" and "marital status" features, since they strongly correlate with sex. See Appendix C.4 for experiments that include "relationship" and "marital-status".

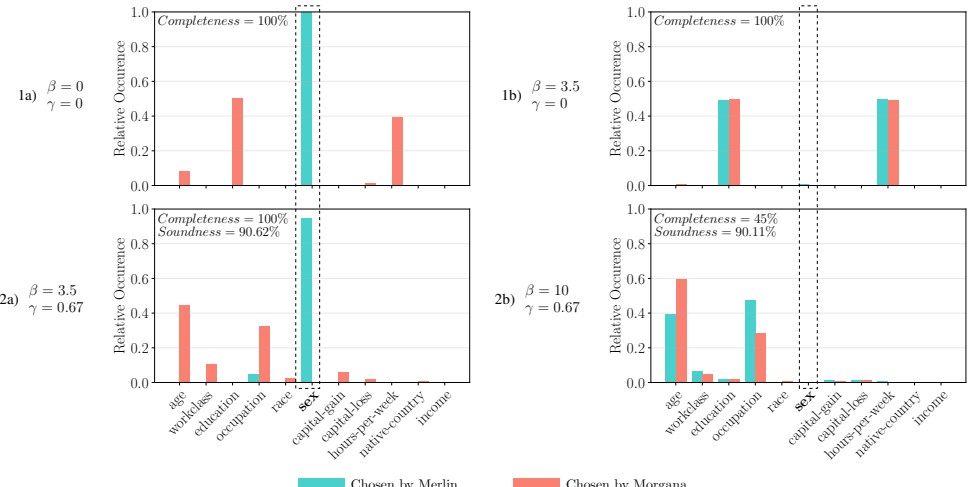

Figure 5: Hiring only men while hiding "sex" as the explanation. 1. No soundness is required ($\gamma = 0$). a) Merlin has no punishment for showing "sex" ($\beta = 0$). He sends Arthur the "sex" feature and they discriminate successfully (high completeness). b) Merlin is incentivised not to use "sex" ($\beta > 0$). He successfully communicates the "sex" to Arthur via different features, here "hours per week" and "education". Morgana can exploit this strategy with the same features switched. 2. High soundness is now required ($\gamma = 0.67$). Merlin either a) shows the "sex" feature despite the punishment ($\beta = 3.5$) and achieves high completeness, or b) hides the "sex" feature ($\beta = 10$) and reduces completeness below 50%, which corresponds to no discrimination.

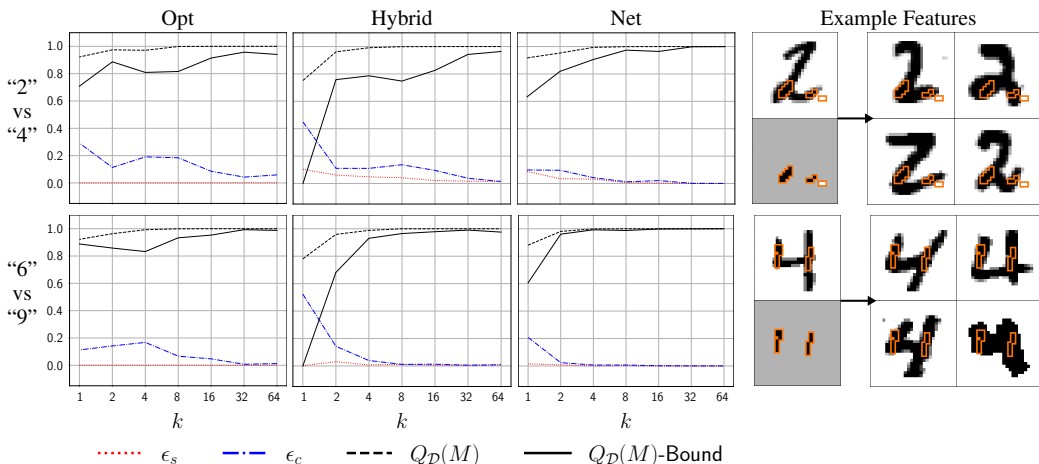

Figure 6: *Left:* For four different setups of Merlin and Morgana, we compare the lower bound on $Q_{\mathcal{D}}(M)$ with the experimental results on the MNIST dataset. The top row is for the labels {"2", "4"}, and the bottom row for {"6", "9"}. The bound is tight for large masks, but loosens sharply for very small mask sizes $k$. *Right:* Examples of the features selected by Merlin for two images. For the "4" feature, there are 13 images in MNIST that share it, all of them of class "4" (we show four here). For the "2" there are 16 images, all of them in "2".

it becomes pessimistic. Regularising Arthur to maintain equal completeness is a potential solution, but this requires global information that for a single batch can only be roughly estimated.

We see in Figure 6, that when Merlin and Morgana are represented by the same method (both optimisers or NNs), the bound is the tightest. In our hybrid approach, Merlin is at a disadvantage, since he needs to learn a general rule on the training set to select good features, whereas Morgana can optimise directly on the test set during evaluation. It appears that the Merlin network is not able to perform as well as the optimiser. In Appendix C, we show error bars sampled over 10 training runs. The training results have greater variance for small masks. This is because there is no optimal strategy for Arthur, who cycles between good completeness and soundness.

## 4  DISCUSSION

It has been shown in Macdonald et al. (2020) that finding informative features for NN classifiers is NP-hard. Thus, any efficient algorithm is a heuristic. Training a Merlin-Arthur classifier is a heuristic as well, as we are not guaranteed to converge to an equilibrium with informative features. However, the framework allows us to verify whether we succeeded by evaluating completeness and soundness on the test dataset. Without it, even just confirming that a given feature is informative is typically an involved task and requires a provably good model of the data distribution Frye et al. (2020); Aas et al. (2021); Wäldchen et al. (2022). An interactive classification setup circumvents these problems. We comment further on this topic in Appendix A.

We can also draw a connection to Adversarial Robustness (Goodfellow et al., 2014). Consider the generation of adversarial examples : $\boldsymbol{\delta}^* = \arg\min_{\|\boldsymbol{\delta}\| \leq \epsilon} L(\mathbf{x} + \boldsymbol{\delta})$, a perturbation that changes the classification but is imperceptible to humans. The intuition behind adversarial robustness is that minuscule changes should not change the classifier decision. Likewise, the intuition behind soundness, i.e., robustness with respect to Morgana, is that hiding parts of an object should not convince the classifier of a different class. At most, one could hide the whole object, which is reflected in the "Don't know!" option. In this sense, soundness is a natural counterpart to adversarial robustness and should be expected of classifiers that generalise to partially hidden objects

### 4.1  IMPACT & LIMITATIONS

Agents with high completeness and soundness can be mandated as commercial classifiers that allow for reliable auditing, e.g., in the context of hiring decisions. Using past decisions by the Merlin-Arthur classifier as ground truth, an auditor uses their own Morgana to verify if the setting has sufficient soundness. If so, the features by Merlin must be the basis of the hiring decisions. The auditor can then inspect them for protected attributes, e.g., race, sex or attributes that strongly correlate with them (Mehrabi et al., 2021). Thus we also see that simply identifying informative features does not fully guarantee that a classifier does not discriminate. For our experiments on the UCI dataset we removed the features "marital status" and "relationship" to make the effect of cheating-prevention more clear. Since they are correlated with sex (thus have high mutual information), they can be used by Merlin to communicate "sex" to Arthur when included, see Appendix C.4. It is then up to social scientists to determine whether the exchanged features constitute discrimination. But this is an issue shared generally by feature-based interpretability approaches.

Our theoretical results are so far only formulated for two classes. We expect them to be extendable to a $n$-class setting, but leave this to further research. It remains to be seen whether our proposed training setup can be extended to more involved datasets while retaining stability. So far, we have kept the training routine straightforward and simple. Similar to GAN-training, which is notoriously difficult (Roth et al., 2017; Wiatrak et al., 2019), there is potential for further sophistication.

## 5  CONCLUSION

We extend the framework of adversarial interactive classifiers to provide quantitative bounds on the mutual information of the selected features. These bounds are in terms of the average precision and ultimately in terms of measurable criteria such as completeness and soundness. We also extend our method beyond the assumption of feature independence that is common in such settings by introducing the notion of Asymmetric Feature Concentration that captures the necessary aspect of the dependence among the features. Finally, we evaluate our results on the UCI Census Income and MNIST datasets and show that the Merlin-Arthur classifier can prevent manipulation that has been demonstrated for other XAI methods and that our theory matches well with our numerical results.

## 6  REPRODUCIBILITY STATEMENT

We have provided all information needed to reproduce the results in the paper and its appendices. Section 3 discusses the various experiments, the datasets used, and provides an overview of the training process. Appendix C reports the experimental setup in more detail and describes the model architecture and the values of the various hyperparameters. We have also provided the source code

we used, which will be available in our Git repository at the time of publication. Proofs of all results stated in the paper are provided in Appendix B.

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

# A  CONCEPTUAL OVERVIEW

As we discuss in Section 4, finding informative features faces two hurdles: a complexity barrier, as well as a modelling problem. Here, we further explain these challenges and how we overcome them. Additionally, we comment further on the related work mentioned in Section 1 and explain why we cannot get guarantees without an adversary present. Furthermore, we show that, with an explicitly constructed data space, the debate setup (Irving et al., 2018) cannot be used to reproduce our results with the same generality. Instead, it would require stronger assumptions on either the classifier or the data space to exclude our counterexample.

## A.1  COMPUTATIONAL COMPLEXITY

Prime Implicant Explanations (Shih et al., 2018), a concept from logical abduction, can be efficiently computed for simple classifiers like decision trees (Izza et al., 2020) and monotonic functions (Marques-Silva et al., 2021). This concept has been extended to NNs (Ignatiev et al., 2019) in the form of *probabilistic* prime implicants, which correspond to features with high precision. However, it has been shown that even approximating small implicants within any non-trivial factor is NP-hard (Waeldchen et al., 2021) for networks of two layers or more. In Blanc et al. (2021), the authors construct an algorithm that circumvents these hardness results by further relaxation of the problem. While this is a noteworthy theoretical breakthrough, the polynomial bound on the feature size grows so quickly with the dimension of the data space that the algorithm does not guarantee useful features for real-world data. For reasonably sized images, one would get guarantees only for features that cover the whole image.

We circumvent the hardness of this problem using a method that is very typical of Deep Learning: Use a heuristic and verify success afterwards! Our approach can be put alongside the regular training of classifiers, which is a theoretically hard problem as well. A heuristic like Stochastic Gradient Descent is not a priori guaranteed to produce a capable classifier. However, we can check the success of the procedure by evaluating the accuracy on a test dataset. In our case, training the Merlin-Arthur classifier is not guaranteed to converge to an equilibrium with informative features. But we can check whether this is the case via the test dataset, where soundness and completeness take the role of the accuracy.

## A.2  MODELLING THE TRUE DATA DISTRIBUTION

We introduce Merlin-Arthur classification as it provides us a way to measure the feature quality via the completeness and soundness values over a test dataset. This would not be necessary if we could directly measure the feature quality over the dataset (though it would still be faster than measuring every individual feature). The reason we need the Merlin-Arthur setup is that for general datasets the conditional entropy

$$H_{\mathbf{y}\sim\mathcal{D}}(c(\mathbf{y}) \,|\, \mathbf{y} \in \phi) = H_{\mathbf{y}\sim\mathcal{D}|_{\mathbf{y}\in\phi}}(c(\mathbf{y})),$$

is difficult to measure, since we do not generally know the conditional distribution $\mathcal{D}|_{\mathbf{y}\in\phi}$. This measurement is possible for MNIST for small features since the dataset is very simple. However, for more complex data, a feature which is large enough to be indicative of the class will in all likelihood not appear more than once in the same dataset. We will now discuss some existing approaches that aim to approximate the conditional data distribution and what problems they face.

Modelling the conditional data distribution has been pursued in the context of calculating Shapley values. These are a different interpretability method based on *characteristic functions* from cooperative game theory that assign a value to every subset of a number of features (Shapley, 2016). We will shortly discuss the approach proposed in Lundberg & Lee (2017), where features correspond to partial vectors supported on sets, see Example 2.2 for how the feature set is defined in that case.

Let $f : [0,1]^d \to \{-1, 1\}$ be a classifier function. Then we can naturally define a characteristic function $\nu_{f,\mathbf{x}} : \mathcal{P}([d]) \to [-1, 1]$ as

$$\nu_{f,\mathbf{x}}(S) = \mathbb{E}_{\mathbf{y}\sim\mathcal{D}}[f(\mathbf{y}) \,|\, \mathbf{y}_S = \mathbf{x}_S] = \int f(\mathbf{y}_S, \mathbf{x}_{S^c}) d\mathbb{P}_{\mathbf{y}\sim\mathcal{D}}(\mathbf{y}_{S^c} \,|\, \mathbf{y}_S = \mathbf{x}_S).$$

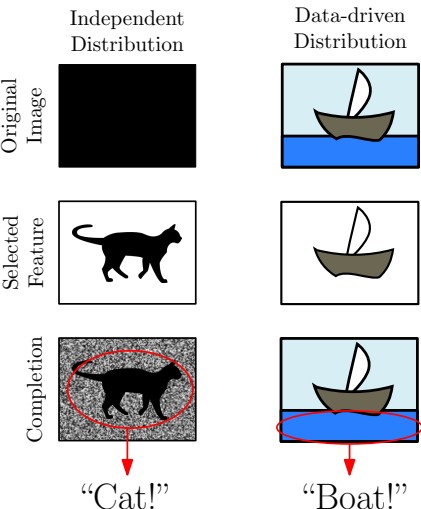

Figure 7: Different failure modes of unrepresentative distributions. *Left:* Independent, random inpainting, similar to Macdonald et al. (2019). From a black image, the shape of a cat is selected, and the rest is filled with uniform noise. The shape of a cat is detected by a classifier. *Right:* Data-driven inpainting, similar to Agarwal & Nguyen (2020). The image of a ship is given and the ship-feature is selected. The data driven distribution inpaints the water back into the image, since in the dataset, ships are always on water. The faulty classifier that relies on the water feature is undetected, as the ship-feature indirectly leads to the correct classification.

The Shapley value for the input component $x_i$ is then defined as

$$\text{Shapley Value}(x_i) = \frac{1}{d} \sum_{S \subseteq [d] \setminus \{i\}} \binom{d-1}{|S|}^{-1} (\nu_{f,\mathbf{x}}(S \cup \{i\}) - \nu_{f,\mathbf{x}}(S)).$$

In Macdonald et al. (2021), the characteristic function is instead used to define a feature selection method as

$$\mathbf{x}_{S^*} \quad \text{where} \quad S^* = \underset{|S| \leq k}{\arg \min} \ \text{dist}(\nu([d]), \nu(S)),$$

where $k \in [d]$ is a cap on the set size and dist is an appropriate distance measure.

As in our setup, the problem is that these approaches depend on how well the conditional probability $\mathbb{P}_{\mathbf{y} \sim \mathcal{D}}(\mathbf{y}_{S^c} \mid \mathbf{y}_S = \mathbf{x}_S)$ is modelled. Modelling the data distribution incorrectly makes it possible to manipulate many existing XAI-methods. This is done by changing the classifier in such a way that it gives the same value on-manifold, but arbitrary values off-manifold. To get feature-based explanations independent of the off-manifold behaviour, one needs to model the data manifold very precisely (Aas et al., 2021; Dombrowski et al., 2019). The authors of Anders et al. (2020); Heo et al. (2019) and Dombrowski et al. (2019) demonstrate this effect for existing techniques, such as sensitivity analysis, LRP, Grad-Cam, IntegratedGradients and Guided Backprop. They are able to manipulate relevance scores at will and demonstrate how this can be used to obfuscate discrimination inside a model. LIME and SHAP can be manipulated as well (Slack et al., 2020) by using a classifier that behaves differently outside off-distribution if the wrong distribution for the explanation. For RDE (Macdonald et al., 2019) it is assumed that features are independent and normally distributed, and it was demonstrated that the off-manifold optimisation can create new features that weren't in the original image (Wäldchen et al., 2022).

We now discuss two approaches proposed to model the data distribution and why each leads to a different problem by under- or over-representing correlation in the data respectively.

**Independent distribution:** Which means that the conditional probability is modelled as

$$\mathbb{P}_{\mathbf{y} \sim \mathcal{D}}(\mathbf{y}_{S^c} \mid \mathbf{y}_S = \mathbf{x}_S) = \prod_{i \in S^c} p(y_i),$$

where $p(y_i)$ are suitable probability densities on the individual input components. This approach has been used in Fong & Vedaldi (2017) and Macdonald et al. (2021), where optimisers are employed to find small features that maximise the classifier score. In fact, Macdonald et al. (2021) has to make a new approximation of the data distribution in every layer and it has been shown that for neural networks one cannot do much better than applying either this or sampling Macdonald & Wäldchen (2022). It was highlighted in Wäldchen et al. (2022) how this approach, when modelling the data distribution incorrectly, will create artificial new features that were not present in the original image. Employing an optimisation method with this distribution can result in masks that generate new features that were not present in the original image. We illustrate this problem in Figure 7. Cutting a specific shape out of a monochrome background will with high likelihood result in an image where this shape is visible. If the distribution was true, a monochrome shape would likely lead to an inpainting that is monochrome in the same colour, destroying the artificial feature. But an independent distribution under-represents these reasonable correlations.

**Taking a data-determined distribution via generative model:**   Which means that the conditional probability is modelled as

$$\mathbb{P}_{\mathbf{y}\sim\mathcal{D}}(\mathbf{y}_{S^c} \mid \mathbf{y}_S = \mathbf{x}_S) = G(\mathbf{y}_{S^c}; \mathbf{x}_S),$$

where $G$ is a suitable generative model. Generative models as a means to approximate the data distribution in the context of explainability have been proposed in a series of publications (Agarwal & Nguyen, 2020; Chang et al., 2018; Liu et al., 2019; Mertes et al., 2020). This setup introduces a problem. If the network and the generator were trained on the same dataset, the biases learned by the classifier will appear might be learned by the generator as well (see Figure 7 for an illustration)! The important cases will be exactly the kind of cases that we will not be able to detect. If the generator has learned that horses and image source tags are highly correlated, it will inpaint an image source tag when a horse is present. This allows the network to classify correctly, even when the network only looks for the tag and has no idea about horses. The faulty distribution over-represents correlations that are not present in the real-world data distribution.

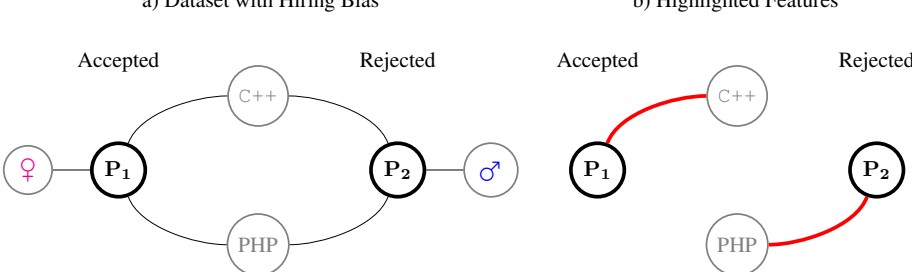

Figure 8: Conceptual illustration of a biased dataset regarding a hiring process and a feature selector that hides this bias. a) Two candidates applied for a position, both are proficient in `C++` and PHP and only differ by sex. The female candidate was accepted, while the male candidate was rejected. b) A feature selector highlights one feature for each candidate. In a *faithfulness* test (Jain et al., 2020), one would retrain a model to reproduce the classification given the selected features to check how much information they carry about the decision. It is clear from this example that the selected features indeed carry this information, not in themselves, but rather through the way they have been selected.

A.3   FEATURE IMPORTANCE AND FAITHFULNESS

Recent work has established *faithfulness* as a reasonable criterion (among others) for sound feature selectors and interpretability approaches (Jain et al., 2020; Rajagopal et al., 2021; Yu et al., 2019). Faithfulness is defined as the ability of the model, retrained solely on the selected features for each data point, to infer the correct classification[2]. We now want to argue how this approach corresponds to our cooperative setup without Morgana and why one must be careful with the conclusions.

---

[2]The setup in Yu et al. (2019) includes an adversary, however, this adversary operates differently from our Morgana and instead tries to classify on the features *not* selected by Merlin. We have not investigated whether this setup is sufficient to prevent hiding biases, as in our example in Figure 8.

If this retraining should fail, it stands to reason that the selected features were not indicative of the class. However, if the retraining succeeds, it is not necessarily true that the features were also used to make the classification. We already commented on the difference between

$$H(c(\mathbf{x})|M(\mathbf{x}) = \phi) \quad \text{and} \quad H(c(\mathbf{y})|\mathbf{y} \in \phi),$$

in Section 3. The retraining corresponds to our purely cooperative setup in the sense that the model can learn in what pattern the feature selector selects features for each class, without the feature itself being indicative, see Figure 8 for an illustration, where a feature selector chooses the feature "proficient in C++" for the accepted candidate and "proficient in PHP" for the rejected one even though both have both features. An adversary could easily exploit these uninformative features by reversing the pattern and fooling the classifier into giving the wrong class. This is why robustness with respect to Morgana is necessary to link $H(c(\mathbf{y})|\mathbf{y} \in \phi)$ and $H(c(\mathbf{x})|M(\mathbf{x}) = \phi)$.

## A.4   DEBATE MODEL VS MERLIN-ARTHUR CLASSIFICATION

The debate setting introduced in Irving et al. (2018) is an intriguing alternative to our proposed setup. However, we are now going to present an example data space on which, in debate mode, Arthur and Merlin can cooperate perfectly without using informative features. For this, we use the fact that in the debate setting, Arthur receives features from both Merlin and Morgana for each classification. Our example illustrates that the debate setting would need stronger requirements on either the data space or Arthur to produce results similar to ours.

Consider the following example of a data space $\mathfrak{D}^{\text{ex}}$, illustrated in Figure 9.

**Example A.1.** *We define the data space* $\mathfrak{D}^{\text{ex}} = ((D, \sigma, \mathcal{D}), c, \Sigma)$ *with*

- $D = \{x_j\}_{j \in [N]}$,
- $\sigma = \mathcal{P}(D)$,
- *for* $T \in \mathcal{P}(D) : \mathcal{D}(T) = \frac{|T|}{N}$,
- $c(x_j) = \begin{cases} -1 & j \text{ odd,} \\ 1 & j \text{ even,} \end{cases}$
- $\Sigma = \{\varnothing\} \cup \bigcup_{j \in [N]} \{\phi_j\}$ *where* $\phi_j = \{x_j, x_{j+1 \bmod N}\}$.

None of the features in $\Sigma$ are informative of the class and the mutual information $I(c(x); x \in \phi)$ for any $\phi \in \Sigma$ is zero. Nevertheless, in a debate setting, Arthur can use the following strategy after receiving a total of two features from Merlin and Morgana

$$A(\{\phi_1, \phi_2\}) = \begin{cases} 0 & \text{if} \phi_1 \cap \phi_2 = \varnothing, \\ \max_j c(x_j) \text{ s.t. } x_j \in \phi_1 \cap \phi_2 & \text{otherwise.} \end{cases}$$

This means he returns the class of the data point with the highest $j$-index that fits the features he is presented (we assume he ignores the $\varnothing$ as a feature). The maximum over the indices should be understood as cyclic, so her prefers $N$ to $N-1$, but 1 to $N$, since $N \mod N = 0$. But now Merlin can use the strategy

$$M(x) = \min_j \phi_j \text{ s.t. } x \in \phi_j,$$

which returns the feature with the lowest $j$-index. It is easy to verify that no matter what Morgana puts forward, a feature with higher or lower index, nothing can convince Arthur of the wrong class. If she gives the same feature as Merlin, the data point will be correctly determined by Arthur. If she gives the other feature, the true data point is the unique one that has both features. Arthur's strategy works as long as *someone* gives him the feature with the lower index.

In a setting where Arthur has to evaluate every feature individually, the best strategy that Arthur and Merlin can use achieves $\epsilon_c = \epsilon_s = \frac{1}{3}$, by making use of the asymmetric feature concentration. The AFC for $\mathfrak{D}^{\text{ex}}$ is $\kappa = 2$, as can be easily checked by taking $F = \{\phi_1, \phi_2, \phi_5, \phi_6\}$ in the definition of the AFC, see Definition B.1, and observing that they cover 4 data points in class $l = -1$ and only two in class $l = 1$. But since the AFC-constant appears in the bound, the lower bound for $Q_M$ is $\frac{1}{6}$, well below the actual average precision of $\frac{1}{2}$.

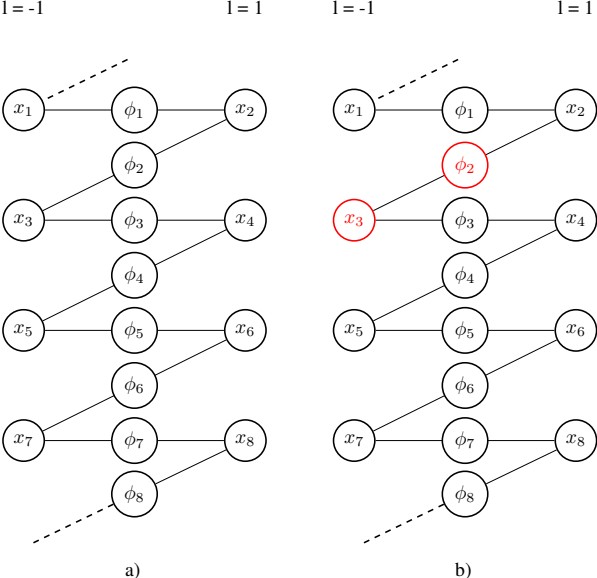

Figure 9: Schematic of $\mathfrak{D}^{\text{ex}}$ as defined in Example A.1. a) The data space forms a tri-partite graph, where every data point shares exactly one feature each with two data points from the opposite class. b) Classification on data point $x_3$. Merlin chooses the feature with the smallest index from this data point, so $\phi_2$. Arthur chooses the class of the data point with the highest index compatible with the presented features. Morgana can choose $\varnothing$, $\phi_2$ or $\phi_3$, but in all cases Arthur can correctly identify the original data point and return class $l = -1$.

This example demonstrates that Arthur and Merlin can successfully cooperate even with uninformative features, as long as Arthur does not have to classify on features by Morgana alone. This implies that to produce similar bounds as in our setup, the debate mode needs stronger restrictions on either the allowed strategies of Arthur or the structure of the data space, such that this example is excluded.

## A.5 DESIGN OF THE THREE-WAY GAME

The basic setup for a prover-verifier game for classification was proposed by Chang et al. with a verifier, a cooperative prover and an adversarial prover for one specific class. The verifier either accepts the evidence for the class or rejects it. Both provers try to convince the verifier, the cooperative prover operates on data from the class, the adversary on data from outside the class. The authors suggest that the way to scale to multiple classes is to have three agents for every class.

In our work, we combine the agents over all classes, to have a single verifier (Arthur), cooperator (Merlin), and adversary (Morgana). The verifier rejecting all classes corresponds to our "Don't know!" option. In our design in Section 3, we make the implicit assumption that the class of the datapoint is unique. Combining the verifiers gives us a numerical advantage for two reasons. First, since a lot of lower-level concepts (e.g. edges and corners for image data) are shared over classes, the lower levels of the neural network benefit by being trained on more and more diverse data. Second, we can leverage the knowledge that the class is unique by outputting a distribution over classes (and "Don't know!"). Both lines of reasoning are standard for deep learning Bridle (1989).

Anil et al. further combine Merlin and Morgana into a single prover that probabilistically produces a certificate for a random class. This has the advantage that it allows for further weight-sharing among the provers. However, the probabilistic nature of the certificate is also a disadvantage. The probability of generating the certificate for the correct class is the inverse of the total number of classes. When applied after training, one only occasionally gets a valid classification. In our case, we can always use Merlin together with Arthur to obtain the correct class together with an interpretable feature.

# B  THEORETICAL DETAILS

We now give further explanations to our theoretical investigation in the main part as well as provide definitions and proofs for the previously stated theorems and lemmas.

## B.1  CONDITIONAL ENTROPY AND AVERAGE PRECISION

We restate the definition of the average precision and average class conditional entropy to show how one can be bound by the other. The average precision of a feature selector $M$ is defined as

$$Q_{\mathcal{D}}(M) := \mathbb{E}_{\mathbf{x} \sim \mathcal{D}}[\mathbb{P}_{\mathbf{y} \sim \mathcal{D}}[c(\mathbf{y}) = c(\mathbf{x}) \,|\, \mathbf{y} \in M(\mathbf{x})]].$$

The average class conditional entropy with respect to a feature selector is defined as

$$H_{\mathbf{x},\mathbf{y} \sim \mathcal{D}}(c(\mathbf{y}) \,|\, \mathbf{y} \in M(\mathbf{x})) := \mathbb{E}_{\mathbf{x} \sim \mathcal{D}}[H_{\mathbf{y} \sim \mathcal{D}}(c(\mathbf{y}) \,|\, \mathbf{y} \in M(\mathbf{x}))].$$

We can expand the latter and reorder that expression in the following way:

$$H_{\mathbf{x},\mathbf{y} \sim \mathcal{D}}(c(\mathbf{y}) \,|\, \mathbf{y} \in M(\mathbf{x})) = -\mathbb{E}_{\mathbf{x} \sim \mathcal{D}}\left[ \sum_{l \in \{-1,1\}} P(c(\mathbf{y}) = l \,|\, \mathbf{y} \in M(\mathbf{x})) \log(P_{\mathbf{y} \sim \mathcal{D}}(c(\mathbf{y}) = l \,|\, \mathbf{y} \in M(\mathbf{x}))) \right]$$

$$= -\mathbb{E}_{\mathbf{x} \sim \mathcal{D}}\left[ \sum_{l \in \{c(\mathbf{x}),-c(\mathbf{x})\}} P(c(\mathbf{y}) = l \,|\, \mathbf{y} \in M(\mathbf{x})) \log(P_{\mathbf{y} \sim \mathcal{D}}(c(\mathbf{y}) = l \,|\, \mathbf{y} \in M(\mathbf{x}))) \right]$$

$$= \mathbb{E}_{\mathbf{x} \sim \mathcal{D}}[H_b(P_{\mathbf{y} \sim \mathcal{D}}(c(\mathbf{y}) = c(\mathbf{x}) \,|\, \mathbf{y} \in M(\mathbf{x}))],$$

where $H_b(p) = -p\log(p) - (1-p)\log(1-p)$ is the binary entropy function. Since $H_b(p)$ is a concave function, we can use Jensen's inequality and arrive at the bound

$$H_{\mathbf{x},\mathbf{y} \sim \mathcal{D}}(c(\mathbf{y}) \,|\, \mathbf{y} \in M(\mathbf{x})) \leq H_b(Q_{\mathcal{D}}(M)).$$

We now give a short proof for Lemma 2.7.

**Lemma 2.7.** *Given $\mathfrak{D} = ((D, \sigma, \mathcal{D}), c, \Sigma)$, a feature selector $M \in \mathcal{M}(D)$ and $\delta \in [0, 1]$. Let $\mathbf{x} \sim \mathcal{D}$, then with probability $1 - \delta^{-1}(1 - Q_{\mathcal{D}}(M))$, $M(\mathbf{x})$ selects a feature $\phi$, s.t.*

$$\mathbb{P}_{\mathbf{y} \sim \mathcal{D}}[c(\mathbf{y}) = c(\mathbf{x}) \,|\, \mathbf{y} \in \phi] \geq 1 - \delta.$$

*Proof.* The proof follows directly from Markov's inequality, which states that for a non-negative random variable $Z$ and $\delta > 0$

$$\mathbb{P}[Z \geq \delta] \leq \frac{\mathbb{E}[Z]}{\delta}.$$

Choosing $Z = 1 - \mathbb{P}_{\mathbf{y} \sim \mathcal{D}}[c(\mathbf{y}) = c(\mathbf{x}) \,|\, \mathbf{y} \in M(\mathbf{x})]$ with $\mathbf{x} \sim \mathcal{D}$ leads to the result. $\square$

## B.2  MIN-MAX THEOREM

We now present the proof for Theorem 2.8 which we restate here.

**Theorem 2.8.** *[Min-Max] Let $M \in \mathcal{M}(D)$ be a feature selector and let*

$$\epsilon_M = \min_{A \in \mathcal{A}} \max_{\widehat{M} \in \mathcal{M}} \mathbb{P}_{\mathbf{x} \sim \mathcal{D}}\left[\mathbf{x} \in E_{M,\widehat{M},A}\right].$$

*Then there exists a set $D' \subset D$ with $\mathbb{P}_{\mathbf{x} \sim \mathcal{D}}[\mathbf{x} \in D'] \geq 1 - \epsilon_M$ such that for $\mathcal{D}' = \mathcal{D}|_{D'}$ we have*

$$Q_{\mathcal{D}'}(M) = 1 \quad \text{and thus} \quad H_{\mathbf{x},\mathbf{y} \sim \mathcal{D}'}(c(\mathbf{y}) \,|\, \mathbf{y} \in M(\mathbf{x})) = 0.$$

*Proof.* From the definition of $\epsilon_M$ it follows that there exists a not necessarily unique $A^\sharp \in \mathcal{A}$ such that

$$\max_{\widehat{M} \in \mathcal{M}} \mathbb{P}_{\mathbf{x} \sim \mathcal{D}}\left[\mathbf{x} \in E_{M,\widehat{M},A^\sharp}\right] = \epsilon_M. \tag{4}$$

Given $A^\sharp$, an optimal strategy by Morgana is certainly

$$\widehat{M^\sharp}(\mathbf{x}) = \begin{cases} \phi \text{ s.t. } A(\phi) = -c(\mathbf{x}) & \text{if possible,} \\ \varnothing & \text{otherwise,} \end{cases}$$

and every optimal strategy differs only on a set of measure zero. Thus, we can define

$$D' = D \setminus E_{M,\widehat{M^\sharp},A^\sharp},$$

and have $\mathbb{P}_{\mathbf{x}\sim\mathcal{D}}[\mathbf{x} \in D'] \geq 1 - \epsilon_M$. We know that $A(M(\mathbf{x})) \neq 0$ when $\mathbf{x} \in D'$ and thus can finally assert that

$$\forall \mathbf{x}, \mathbf{y} \in D' : \mathbf{y} \in M(\mathbf{x}) \Rightarrow c(\mathbf{y}) = c(\mathbf{x}).$$

Otherwise there would be at least one $\mathbf{y} \in D'$ that would also be in $E_{M,\widehat{M^\sharp},A^\sharp}$. Thus, we conclude $Q_{\mathcal{D}'}(M) = 1$, and from

$$0 \leq H_{\mathbf{x},\mathbf{y}\sim\mathcal{D}'}(c(\mathbf{y}) \mid \mathbf{y} \in M(\mathbf{x})) \leq H_b(Q_{\mathcal{D}'}(M)) = 0,$$

it follows that $H_{\mathbf{x},\mathbf{y}\sim\mathcal{D}'}(c(\mathbf{y}) \mid \mathbf{y} \in M(\mathbf{x})) = 0$. $\qquad\square$

This theorem states that if Merlin uses a strategy that allows Arthur to classify almost always correctly, thus small $\epsilon_M$, then there exists a dataset that covers almost the entire original dataset and on which the class entropy conditioned on the features selected by Merlin is zero.

This statement with a new set $D'$ appears convoluted at first, and we would prefer a simple bound, such as

$$Q_{\mathcal{D}}(M) \geq 1 - \epsilon_M,$$

where we take the average precision over the whole dataset. This is, however, not straightforwardly possible due to a principle we call *asymmetric feature concentration* and which we introduce in the next section.

### B.3 ASYMMETRIC FEATURE CONCENTRATION

*Asymmetric feature concentration (AFC)* is a concept that will allow us to state our main result. It measures if there is a set of features that are concentrated in a few data points in one class, but spread out over almost all data points in the other class. This represents a possible quirk of datasets that can result in a scenario where Arthur and Merlin cooperate successfully with high probability, Morgana is unable to fool Arthur except with low probability– and yet the features exchanged by Arthur and Merlin are uninformative for the class. An illustration of such an unusual dataset is given in Figure 10.

For an illustration of the asymmetric feature concentration, consider two-class data space $\mathfrak{D} = \{(D, \sigma, \mathcal{D}), c, \Sigma\}$, e.g. the "fish and fruit" data depicted in Figure 10. Let us choose $F \subset \Sigma$ to be all the "fish" features. We see that these features are strongly anti-concentrated in class $l = -1$ (none of them share an image) and strongly concentrated in class $l = 1$ (all of them are in the same image).

For now, let us assume $F$ is finite and let $\mathcal{F} = \mathcal{U}(F)$, the uniform measure over $F$. We have strong AFC if the class-wise ratio of what each feature covers individually is much larger than what the features cover as a whole:

$$\mathbb{E}_{\phi\sim\mathcal{F}}\left[\frac{\mathbb{P}_{\mathbf{x}\sim\mathcal{D}_{-l}}[\mathbf{x} \in \phi]}{\mathbb{P}_{\mathbf{x}\sim\mathcal{D}_l}[\mathbf{x} \in \phi]}\right] \gg \frac{\mathbb{P}_{\mathbf{x}\sim\mathcal{D}_{-l}}[\mathbf{x} \in \cup F]}{\mathbb{P}_{\mathbf{x}\sim\mathcal{D}_l}[\mathbf{x} \in \cup F]}.$$

In our example, every individual "fish" feature covers one image in each class, so the left side is equal to 1. As a feature set, they cover 6 images in class $-1$ and one in class $1$, so the right side is $\frac{1}{6}$. Using

$$\frac{\mathbb{P}[\mathbf{x} \in \phi]}{\mathbb{P}[\mathbf{x} \in \cup F]} = \mathbb{P}[\mathbf{x} \in \phi \mid \mathbf{x} \in \cup F],$$

we can restate this expression as

$$\mathbb{E}_{\phi\sim\mathcal{F}}[\kappa_l(\phi, F)] \gg 1, \tag{5}$$

where

$$\kappa_l(\phi, F) = \frac{\mathbb{P}_{\mathbf{x}\sim\mathcal{D}_{-l}}[\mathbf{x} \in \phi \mid \mathbf{x} \in \cup F]}{\mathbb{P}_{\mathbf{x}\sim\mathcal{D}_l}[\mathbf{x} \in \phi \mid \mathbf{x} \in \cup F]}.$$

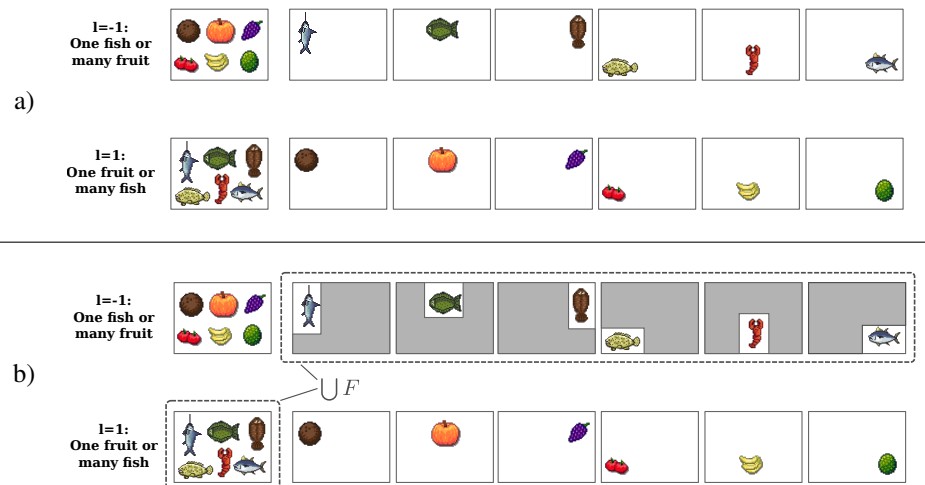

Figure 10: Illustration of a data space with strong asymmetric feature concentration. *a):* A dataset with fish and fruit features. The features are asymmetrically correlated, because all the fruit features are maximally correlated in class $-1$ (they are all in the same image) and maximally uncorrelated in 1 (no two fruits share the same image). The reverse is true for the fruits. See Figure 11 for a strategy for Merlin that ensures strong completeness and soundness with uninformative features.

*b):* Asymmetric feature concentration for a specific feature selection. For the class $-1$, we select the set $F$ of all "fish" features. Each individual fish feature in $F$ covers a fraction of $\frac{1}{6}$ of $(\cup F) \cap D_{-1}$ and all images (one) in $(\cup F) \cap D_1$. The expected value in Equation (5) thus evaluates to $k = 6$. This is also the maximum AFC for the entire dataset as no different feature selection gives a higher value.

For our "fish" features we have $\kappa_{-1}(\phi, F) = 6$ for every feature $\phi \in F$. Considering an infinite set $F$, we need a way to get a reasonable measure $\mathcal{F}$, where we don't want to "overemphasise" features that appear only in very few data points. We thus define a feature selector $f_F \in \mathcal{M}(\cup F)$ as

$$f_F(\mathbf{x}) = \underset{\substack{\phi \in F \\ \text{s.t. } \mathbf{x} \in \phi}}{\arg\max} \, \kappa(\phi, F), \tag{6}$$

and we can define the push-forward measure $\mathcal{F} = f_{F*}\mathcal{D}_l|_{\cup F}$. For our fish and fruit example, where $F$ is the set of all fish features, $f_{F*}$ would select a fish feature for every image in class $-1$ that is in $\cup F$. Putting everything together, we get the following definition.

**Definition B.1** (Asymmetric feature concentration). *Let $((D, \sigma, \mathcal{D}), c, \Sigma)$ be a two-class data space, then the asymmetric feature concentration $\kappa$ is defined as*

$$\kappa = \max_{l \in \{-1, 1\}} \max_{F \subset \Sigma} \mathbb{E}_{\mathbf{y} \sim \mathcal{D}_l|_{\cup F}} \left[ \max_{\substack{\phi \in F \\ \text{s.t. } \mathbf{y} \in \phi}} \frac{\mathbb{P}_{\mathbf{x} \sim \mathcal{D}_{-l}}[\mathbf{x} \in \phi \mid \mathbf{x} \in \cup F]}{\mathbb{P}_{\mathbf{x} \sim \mathcal{D}_l}[\mathbf{x} \in \phi \mid \mathbf{x} \in \cup F]} \right].$$

As we have seen in the "fish and fruit" example, the AFC can be made arbitrarily large, as long as one can fit many individual features into a single image. We can prove that the maximum amount of features per data point indeed also gives an upper bound on the AFC. We now come back to Lemma 2.10 and prove it.

**Lemma 2.10.** *Let $\mathfrak{D} = ((D, \sigma, \mathcal{D}), c, \Sigma)$ be a two-class data space with AFC of $\kappa$. Let $K = \max_{\mathbf{x} \in D} |\{\phi \in \Sigma \mid \mathbf{x} \in \phi\}|$ be the maximum number of features per data point. Then $\kappa \leq K$.*

*Proof.* Let $l \in \{-1, 1\}$ and let $F \subset \Sigma$. We define $f_F \in \mathcal{M}(\cup F)$ as in Equation (6) as well as $\mathcal{F} = f_{F*}\mathcal{D}_l|_{\cup F}$. We can assert that

$$\mathbb{P}_{\mathbf{x} \sim \mathcal{D}_l}[\mathbf{x} \in \phi \mid \mathbf{x} \in \cup F] \geq \mathbb{P}_{\mathbf{x} \sim \mathcal{D}_l}[f(\mathbf{x}) = \phi \mid \mathbf{x} \in \cup F] = \mathcal{F}(\phi). \tag{7}$$

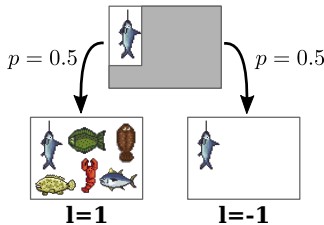

**l=1**        **l=-1**

Figure 11: In the dataset presented in Figure 10, Merlin can use the strategy to always select the fish features for class $l = -1$ and the fruit features for class $l = 1$ if they exist and choose something arbitrary otherwise. Arthur can then guess $l = 1$ if he gets a fish and $l = -1$ for a fruit. This strategy fails only for the images containing all fruits or fish, and can only be exploited by Morgana for those same two images out of 14. The completeness and soundness constants in this case are both $\frac{1}{7}$. But as illustrated here, each "fish" feature is completely uninformative of the class. Conditioned on the selected fish, it could either be the image from class $l = -1$ or from $l = 1$.

We then switch the order of taking the expectation value in the definition of the AFC:

$$\kappa_l(F) = \mathbb{E}_{\phi \sim \mathcal{F}} \left[ \frac{\mathbb{P}_{\mathbf{x} \sim \mathcal{D}_{-l}}[\mathbf{x} \in F \,|\, \mathbf{x} \in \cup F]}{\mathbb{P}_{\mathbf{y} \sim \mathcal{D}_l}[\mathbf{y} \in \phi \,|\, \mathbf{y} \in \cup F]} \right]$$

$$= \mathbb{E}_{\phi \sim \mathcal{F}} \left[ \frac{\mathbb{E}_{\mathbf{x} \sim \mathcal{D}_{-l}}[\mathbb{1}(\mathbf{x} \in \phi) \,|\, \mathbf{x} \in \cup F]}{\mathbb{P}_{\mathbf{y} \sim \mathcal{D}_l}[\mathbf{y} \in \phi \,|\, \mathbf{y} \in \cup F]} \right]$$

$$= \mathbb{E}_{\mathbf{x} \sim \mathcal{D}_{-l}} \left[ \mathbb{E}_{\phi \sim \mathcal{F}} \left[ \frac{\mathbb{1}(\mathbf{x} \in \phi)}{\mathbb{P}_{\mathbf{y} \sim \mathcal{D}_l}[\mathbf{y} \in \phi \,|\, \mathbf{y} \in \cup F]} \right] \,\Big|\, \mathbf{x} \in \cup F \right].$$

Since there are only finitely many features in a data point we can express the expectation value over a countable sum weighted by the probability of each feature:

$$\kappa_l(F) = \mathbb{E}_{\mathbf{x} \sim \mathcal{D}_{-l}} \left[ \sum_{\phi \in F \,:\, \mathbf{x} \in F} \left[ \frac{\mathcal{F}(\phi)}{\mathbb{P}_{\mathbf{y} \sim \mathcal{D}_l}[\mathbf{y} \in \phi \,|\, \mathbf{y} \in \cup F]} \right] \,\Big|\, \mathbf{x} \in \cup F \right]$$

$$\leq \mathbb{E}_{\mathbf{x} \sim \mathcal{D}_{-l}} \left[ \sum_{\phi \in F \,:\, \mathbf{x} \in F} 1 \,\Big|\, \mathbf{x} \in \cup F \right]$$

$$\leq \mathbb{E}_{\mathbf{x} \sim \mathcal{D}_{-l}}[K \,|\, \mathbf{x} \in \cup F]$$

$$= K,$$

where in the first step we used Equation (7) and the definition of $K$ in the second. Then we see that

$$\kappa = \max_{l \in \{-1, 1\}} \max_{F \subset \Sigma} \kappa_l(F) \leq K.$$

$\square$

The number of features per data point is dependent on which kinds of features we consider. Without limitations, this number can be $2^d$, i.e., exponentially high. See Figure 12 for an example of exponentially large AFC parameters. If we consider only features of a fixed size and shape, such as in image data, the number of features per data point drops to $\approx d$.

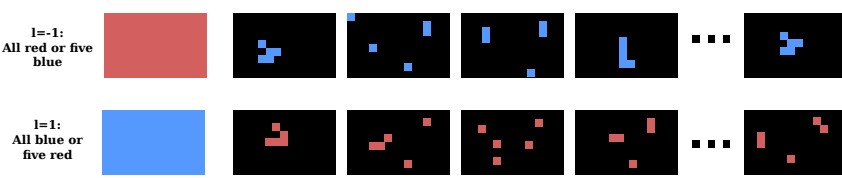

Figure 12: An example of a dataset with very high asymmetric feature concentration. The completely red image shares a feature with each of the $m$-red-pixel images (here $m = 5$), of which there are $\binom{d}{m}$ many. In the worst case $m = \frac{d}{2}$, resulting in $k = \binom{d}{d/2}$ thus exponential growth in $d$.

We now prove an intermediate lemma that will later allow us to prove Theorem 2.12.

**Lemma B.2.** *Let $((D, \sigma, \mathcal{D}), c, \Sigma)$ be a two-class data space with asymmetric feature concentration of $\kappa$ and class imbalance $B$. Let $A : [0,1]^d \to \{-1, 0, 1\}$ be a feature classifier and $M \in \mathcal{M}(D)$ a feature selector for $D$. From*

*1. Completeness:*

$$\min_{l \in \{-1,1\}} \mathbb{P}_{\mathbf{x} \sim \mathcal{D}_l}[A(M(\mathbf{x})) = l] \geq 1 - \epsilon_c,$$

*2. Soundness:*

$$\max_{\widehat{M} \in \mathcal{M}(D)} \max_{l \in \{-1,1\}} \mathbb{P}_{\mathbf{x} \sim \mathcal{D}_l}\left[A\left(\widehat{M}(\mathbf{x})\right) = -l\right] \leq \epsilon_s,$$

*follows*

$$Q_{\mathcal{D}}(M) \geq 1 - \epsilon_c - \frac{\kappa \epsilon_s}{1 - \epsilon_c + \kappa \epsilon_s B^{-1}}.$$

This lemma gives a bound on the probability that data points with features selected by Merlin will also belong to the same class. This probability is large as long as we have a bound on the AFC of the dataset.

*Proof.* The proof of our lemma is fairly intuitive, although the notation can appear cumbersome. Here we give a quick overview over the proof steps.

1. In the definition of the AFC, we maximise over all possible features sets. We will choose as a special case (for each class $l \in \{-1, 1\}$) the features that Merlin selects for data points that Arthur classifies correctly.

2. These feature sets cover the origin class at least up to $1 - \epsilon_c$, and the other class at most up to $\epsilon_s$, which is required by the completeness and soundness criteria, respectively. This gives us a high precision for the whole feature set.

3. The AFC upper bounds the quotient of the precision of the whole feature set and expected precision of the individual features, which finally allows us to state our result.

Let us define a partition of $D$ according to the true class and the class assigned by Arthur. From now on, let $l \in \{-1, 1\}$ and $m \in \{-1, 0, 1\}$. We introduce the datasets

$$D_{l,m} = \{\mathbf{x} \in D_l \mid A(M(\mathbf{x})) = m\},$$

which means that $D_{l,l}$ are the data points that Arthur classifies correctly, and furthermore

$$F_l = M(D_{l,l}).$$

To use the AFC bound we need a feature selector $f : D_l|_{\cup F} \to F$. Merlin itself maps to features outside $F$ when applied to data points in $D_l|_{\cup F_l} \setminus D_{l,l}$. Let us thus define $\sigma : D_F \setminus D_{l,l} \to F$ which selects an arbitrary feature from $F$ (in case one is concerned whether such a representative can always be chosen, consider a discretised version of the data space which allows for an ordering). Then we can define

$$f(\mathbf{x}) = \begin{cases} M(\mathbf{x}) & \mathbf{x} \in D_{l,l}, \\ \sigma(\mathbf{x}) & \mathbf{x} \in D_l|_{\cup F_l} \setminus D_{l,l}, \end{cases} \quad \text{and} \quad \mathcal{F}_l = f_* D_l|_{\cup F_l}.$$

This feature selector will allow us to use the AFC bound. We now abbreviate

$$p(\mathbf{x}, f) = \mathbb{P}_{\mathbf{y} \sim \mathcal{D}}[c(\mathbf{y}) \neq c(\mathbf{x}) \mid \mathbf{y} \in f(\mathbf{x})] \quad \text{and} \quad P_l = \mathbb{P}_{\mathbf{x} \sim \mathcal{D}}[\mathbf{x} \in D_l].$$

Then

$$1 - Q_{\mathcal{D}}(M) = \mathbb{E}_{\mathbf{x} \sim \mathcal{D}}[p(\mathbf{x}, M)] = \sum_{l \in \{-1,1\}} \mathbb{E}_{\mathbf{x} \sim \mathcal{D}_l}[p(\mathbf{x}, M)] P_l. \tag{8}$$

Using the completeness criterion and the fact that $p(\mathbf{x}, M) \leq 1$ we can bound

$$
\begin{aligned}
\mathbb{E}_{\mathbf{x}\sim\mathcal{D}_l}[p(\mathbf{x}, M)] &= \mathbb{E}_{\mathbf{x}\sim\mathcal{D}_l}[p(\mathbf{x}, M)\mathbb{1}(\mathbf{x} \in D_{l,l})] + \mathbb{E}_{\mathbf{x}\sim\mathcal{D}_l}[p(\mathbf{x}, M)\mathbb{1}(\mathbf{x} \notin D_{l,l})] \\
&\leq \mathbb{E}_{\mathbf{x}\sim\mathcal{D}_l}[p(\mathbf{x}, M)\mathbb{1}(\mathbf{x} \in D_{l,l})] + \epsilon_c \\
&\leq \mathbb{E}_{\mathbf{x}\sim\mathcal{D}_l}[p(\mathbf{x}, M)\mathbb{1}(\mathbf{x} \in D_{l,l})] + \epsilon_c + \mathbb{E}_{\mathbf{x}\sim\mathcal{D}_l}[p(\mathbf{x}, \sigma)\mathbb{1}(\mathbf{x} \in D_l|_{\cup F_l} \setminus D_{l,l})] \\
&\leq \frac{\mathbb{E}_{\mathbf{x}\sim\mathcal{D}_l}[p(\mathbf{x}, M)\mathbb{1}(\mathbf{x} \in D_{l,l}) + p(\mathbf{x}, \sigma)\mathbb{1}(\mathbf{x} \in D_l|_{\cup F_l} \setminus D_{l,l})]}{\mathbb{P}_{\mathbf{x}\sim\mathcal{D}_l}[\mathbf{x} \in D_l|_{\cup F_l}]} + \epsilon_c \\
&= \mathbb{E}_{\mathbf{x}\sim\mathcal{D}_l|_{\cup F_l}}[p(\mathbf{x}, f)] + \epsilon_c \\
&= \mathbb{E}_{\phi\sim\mathcal{F}_l}[\mathbb{P}_{\mathbf{y}\sim\mathcal{D}}[c(\mathbf{y}) = -l \mid \mathbf{y} \in \phi]] + \epsilon_c.
\end{aligned}
$$

We can expand the expression in the expectation using Bayes' Theorem:

$$
\begin{aligned}
\mathbb{P}_{\mathbf{y}\sim\mathcal{D}}[c(\mathbf{y}) = -l \mid \mathbf{y} \in \phi] &= \frac{\mathbb{P}_{\mathbf{y}\sim\mathcal{D}_{-l}}[\mathbf{y} \in \phi]\mathbb{P}_{-l}}{\mathbb{P}_{\mathbf{y}\sim\mathcal{D}_{-l}}[\mathbf{y} \in \phi]P_{-l} + \mathbb{P}_{\mathbf{y}\sim\mathcal{D}_l}[\mathbf{y} \in \phi]P_l} \\
&= h\left(\frac{\mathbb{P}_{\mathbf{y}\sim\mathcal{D}_{-l}}[\mathbf{y} \in \phi]\mathbb{P}_{-l}}{\mathbb{P}_{\mathbf{y}\sim\mathcal{D}_l}[\mathbf{y} \in \phi]P_l}\right),
\end{aligned}
$$

where $h(t) = (1+t^{-1})^{-1}$. Since $h(t)$ is a concave function for $t \geq 0$, we know that for any random variable $R$ have $\mathbb{E}[h(R)] \leq h(\mathbb{E}[R])$, so

$$
\mathbb{E}_{\mathbf{x}\sim\mathcal{D}_l}[p(\mathbf{x}, M)] \leq h\left(\mathbb{E}_{\phi\sim\mathcal{F}_l}\left[\frac{\mathbb{P}_{\mathbf{y}\sim\mathcal{D}_{-l}}[\mathbf{y} \in \phi]}{\mathbb{P}_{\mathbf{y}\sim\mathcal{D}_l}[\mathbf{y} \in \phi]}\right]\frac{P_{-l}}{P_l}\right) + \epsilon_c. \tag{9}
$$

From the definition of the AFC $\kappa$ we know that

$$
\mathbb{E}_{\phi\sim\mathcal{F}_l}\left[\frac{\mathbb{P}_{\mathbf{y}\sim\mathcal{D}_{-l}}[\mathbf{y} \in \phi]}{\mathbb{P}_{\mathbf{y}\sim\mathcal{D}_l}[\mathbf{y} \in \phi]}\right] \leq \mathbb{E}_{\mathbf{x}\sim\mathcal{D}_l|_{\cup F_l}}\left[\max_{\substack{\phi\in F \\ \text{s.t. } \mathbf{x}\in\phi}} \frac{\mathbb{P}_{\mathbf{y}\sim\mathcal{D}_{-l}}[\mathbf{y} \in \phi]}{\mathbb{P}_{\mathbf{y}\sim\mathcal{D}_l}[\mathbf{y} \in \phi]}\right] \leq \kappa\frac{\mathbb{P}_{\mathbf{y}\sim\mathcal{D}_{-l}}[\mathbf{y} \in \cup F]}{\mathbb{P}_{\mathbf{y}\sim\mathcal{D}_l}[\mathbf{y} \in \cup F]}. \tag{10}
$$

Now we make use of the fact that we can lower bound $\mathbb{P}_{\mathbf{y}\sim\mathcal{D}_l}[\mathbf{y} \in F]$ by the completeness property

$$
\mathbb{P}_{\mathbf{y}\sim\mathcal{D}_l}[\mathbf{y} \in \cup F] \geq 1 - \epsilon_c,
$$

and upper bound $\mathbb{P}_{\mathbf{y}\sim\mathcal{D}_{-l}}[\mathbf{y} \in \cup F]$ with the soundness property

$$
\mathbb{P}_{\mathbf{y}\sim\mathcal{D}_{-l}}[\mathbf{y} \in \cup F] \leq \epsilon_s.
$$

This is because $\mathbf{y} \in \cup F$ implies that there are features Morgana can use to convince Arthur of class $l$ whereas $\mathbf{y} \sim \mathcal{D}_{-l}$. Together with Equation (9) and Equation (10) we arrive at

$$
\mathbb{E}_{\mathbf{x}\sim\mathcal{D}_l}[p(\mathbf{x}, M)] \leq h\left(\kappa\frac{\epsilon_s}{1-\epsilon_c}\frac{P_{-l}}{P_l}\right) + \epsilon_c = \frac{\kappa\epsilon_s\frac{P_{-l}}{P_l}}{1 - \epsilon_c + \kappa\epsilon_s\frac{P_{-l}}{P_l}} + \epsilon_c.
$$

Using $\frac{P_l}{P_{-l}} \leq B$ thus $\frac{P_{-l}}{P_l} \geq B^{-1}$, we can express

$$
\mathbb{E}_{\mathbf{x}\sim\mathcal{D}_l}[p(\mathbf{x}, M)] \leq \frac{\kappa\epsilon_s\frac{P_{-l}}{P_l}}{1 - \epsilon_c + \kappa\epsilon_s B^{-1}} + \epsilon_c.
$$

Inserted back into equation Equation (8) leads us to

$$
1 - Q_\mathcal{D}(M) \leq \frac{\kappa\epsilon_s}{1 - \epsilon_c + \kappa\epsilon_s B^{-1}} + \epsilon_c.
$$

$\square$

## B.4 RELATIVE SUCCESS RATE AND REALISTIC ALGORITHMS

As discussed in Section 2.3, Morgana only has to find features that are also discoverable by Merlin—albeit in images of a different class. The question then becomes how much the other features in the image (the context) matter for finding this feature. We can easily construct scenarios in which the context matters very strongly, see Figure 13 for an example. We expect that for most real-world data this dependence is only weak and can be upper bounded.

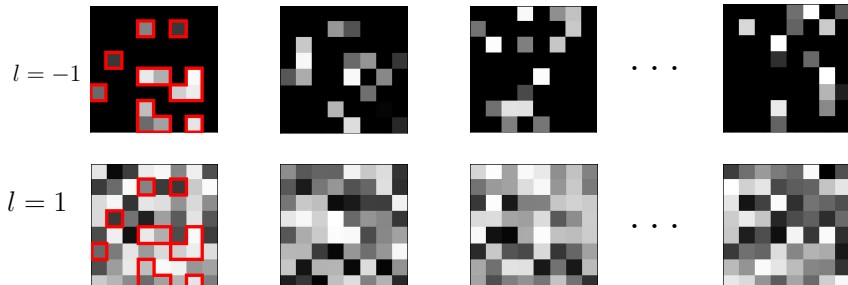

Figure 13: Illustration of a dataset with strong context dependence. Class $-1$ consists of $k$-sparse images whose pixel values sum to some number $S$. For each of these images, there is a non-sparse image in class $1$ that shares all non-zero values (marked in red for the first image). Merlin can use the strategy to show all $k$ non-zero pixels for an image from class $-1$ and $k+1$ arbitrary non-zero pixels for class $1$. Arthur checks if the sum is equal to $S$ or if the number of pixels equal to $k+1$, otherwise he says "I don't know!". He will then classify with 100% accuracy. Nevertheless, the features Merlin uses for class $-1$ are completely uncorrelated with the class label. To exploit this, however, Morgana would have to solve the NP-hard (Kleinberg & Tardos, 2006) subset sum problem to find the pixels for images in class $1$ that sum to $S$. The question is not in which class we can find the features, but in which class we can find the features *efficiently*.

Let us make this notion more formal. We want to compare the success rates of Merlin and Morgana conditioned on a convincing feature $\phi$ with $A(\phi) = l$ being present in the data point:

$$\frac{\mathbb{P}_{\mathbf{x}\sim\mathcal{D}_{-l}}\left[\widehat{M} \text{ succeeds} \mid \mathbf{x} \in \phi\right]}{\mathbb{P}_{\mathbf{x}\sim\mathcal{D}_l}[M \text{ succeeds} \mid \mathbf{x} \in \phi]}.$$

The question is how to define the "success". If we measure success by returning $\phi$, it might be that Morgana can actually find a different feature that convinces Arthur, which in that case would not count as a success. If we instead just require that $A(M(\mathbf{x}, l)) = l$, then Merlin could instead rely on a different, easily findable feature that is only present in data points of the correct class. The feature $\phi$ might then be generally hard to find, and Morgana would need to be again a perfect optimiser. We reason that both approaches would make the fraction unreasonably large even for practical datasets.

The solution is to make the fraction asymmetric and require Merlin to actually find the feature $\phi$ and Morgana to simply convince Arthur on the data point. So we consider the expression

$$\frac{\mathbb{P}_{\mathbf{x}\sim\mathcal{D}_{-l}}\left[A(\widehat{M}(\mathbf{x})) = l \mid \mathbf{x} \in \phi\right]}{\mathbb{P}_{\mathbf{x}\sim\mathcal{D}_l}[M(\mathbf{x}) = \phi \mid \mathbf{x} \in \phi]},$$

and try to assume an upper bound. In the following definition, we apply this idea to whole sets of features on which Arthur is constant. That means we maximise over sets $F \subseteq A^{-1}(l)$ where $A^{-1}$ is the preimage of $A$. If Merlin can find features from this set in one class, then Morgana should not have a much harder time finding them in the other class, given that they are present in the data point.

**Definition B.3** (Relative Success Rate). *Let $\mathfrak{D} = ((D, \sigma, \mathcal{D}), c, \Sigma)$ be a two-class data space. Let $A \in \mathcal{A}$ and $M \colon D \times \{-1, 1\} \to \Sigma^d$ an approximation for $M, \widehat{M} \in \mathcal{M}(D)$. Then the relative success rate $\alpha$ of $\widehat{M}$ with respect to $A, M$ and $\mathfrak{D}$ is defined as*

$$\alpha := \min_{l\in\{-1,1\}} \min_{F\subseteq A^{-1}(l)} \frac{\mathbb{P}_{\mathbf{x}\sim\mathcal{D}_{-l}}\left[A(\widehat{M}(\mathbf{x})) = l \mid \mathbf{x} \in \cup F\right]}{\mathbb{P}_{\mathbf{x}\sim\mathcal{D}_l}[M(\mathbf{x}) \in F \mid \mathbf{x} \in \cup F]}.$$

It might make sense for the definition to require that $\mathcal{D}(\cup F)$ to be large, so that no insignificant feature sets diminish the relative success rate, but since we cannot evaluate this quantity numerically anyway we leave this unrestricted for now.

**Example B.4.** *Let $\mathfrak{D}$ be a data space with maximum number of features per data point $K$. Let Morgana operate the algorithm described in Algorithm 1, in which she randomly searches for a convincing feature. Then we have*

$$\alpha \geq \frac{K}{n_{try}},$$

*which corresponds to an upper bound on the probability that Morgana will succeed on an individual data point when there is only one convincing feature.*

**Algorithm 1** Merlin-Arthur Training

1: **Input:** $\mathbf{x} \in D, n_{\text{try}}$
2: **Output:** $\phi \in \Sigma$
3: **for** $i \in [n_{\text{try}}]$ **do**
4:     Pick random feature $\phi$ s.t. $\mathbf{x} \in \phi$
5:     **if** $A(\phi) = -c(\mathbf{x})$ **then**
6:         **return** $\phi$
7:     **end if**
8: **end for**
9: **return** $\varnothing$

We generally want Morgana's algorithm to be at least as powerful as Merlin's so in case of an optimiser one can consider giving more iterations or more initial starting values.

We now want to prove Theorem 2.12.

**Theorem 2.12.** *Let $\mathfrak{D} = ((D, \sigma, \mathcal{D}), c, \Sigma)$ be a two-class data space with AFC of $\kappa$ and class imbalance $B$. Let $A \in \mathcal{A}$, and $M, \widehat{M} \in \mathcal{M}(D)$ such that $\widehat{M}$ has a relative success rate of $\alpha$ with respect to $A, M$ and $\mathfrak{D}$. Define*

*1. Completeness:* $\quad \min\limits_{l \in \{-1,1\}} \mathbb{P}_{\mathbf{x} \sim \mathcal{D}_l}[A(M(\mathbf{x})) = c(\mathbf{x})] \geq 1 - \epsilon_c,$

*2. Soundness:* $\quad \max\limits_{l \in \{-1,1\}} \mathbb{P}_{\mathbf{x} \sim \mathcal{D}_l}\left[A\left(\widehat{M}(\mathbf{x})\right) = -c(\mathbf{x})\right] \leq \epsilon_s,$

*where $\mathcal{D}_l$ is the data distribution conditioned on the class $l$. Then it follows that*

$$Q_{\mathcal{D}}(M) \geq 1 - \epsilon_c - \frac{\kappa \alpha^{-1} \epsilon_s}{1 - \epsilon_c + \kappa \alpha^{-1} B^{-1} \epsilon_s}.$$

*Proof.* We follow the same proof steps and definitions as in the proof of Lemma B.2 up to Equation (10). Then we consider the following

$$\alpha \frac{\mathbb{P}_{\mathbf{y} \sim \mathcal{D}_{-l}}[\mathbf{y} \in F]}{\mathbb{P}_{\mathbf{y} \sim \mathcal{D}_l}[\mathbf{y} \in F]} \leq \frac{\mathbb{P}_{\mathbf{y} \sim \mathcal{D}_{-l}}[\mathbf{y} \in F]}{\mathbb{P}_{\mathbf{y} \sim \mathcal{D}_l}[\mathbf{y} \in F]} \frac{\mathbb{P}_{\mathbf{x} \sim \mathcal{D}_{-l}}[A(M(\mathbf{x}, l)) = l \mid \mathbf{x} \in F]}{\mathbb{P}_{\mathbf{x} \sim \mathcal{D}_l}[M(\mathbf{x}, l) \in F \mid \mathbf{x} \in F]} \tag{11}$$

$$= \frac{\mathbb{P}_{\mathbf{x} \sim \mathcal{D}_{-l}}[A(M(\mathbf{x}, l)) = l, \mathbf{x} \in F]}{\mathbb{P}_{\mathbf{x} \sim \mathcal{D}_l}[M(\mathbf{x}, l) \in F, \mathbf{x} \in F]} \tag{12}$$

$$= \frac{\mathbb{P}_{\mathbf{x} \sim \mathcal{D}_{-l}}[A(M(\mathbf{x}, l)) = l]}{\mathbb{P}_{\mathbf{x} \sim \mathcal{D}_l}[M(\mathbf{x}, l) \in F]}, \tag{13}$$

where we used in the last step that $M(\mathbf{x}, l) \in F \Rightarrow \mathbf{x} \in F$ and that when $\mathbf{x} \in D_l$ we have $A(M(\mathbf{x}, l)) = l \Rightarrow \mathbf{x} \in F$ both by definition of $F$. We know by the soundness criterion

$$\mathbb{P}_{\mathbf{x} \sim \mathcal{D}_{-l}}[A(M(\mathbf{x}, l)) = l] \leq \epsilon_s.$$

From the definition of $F$ and the completeness criterion we get $\mathbb{P}_{\mathbf{x} \sim \mathcal{D}_l}[M(\mathbf{x}, l) \in F] \geq 1 - \epsilon_c$. Putting everything together we arrive at

$$\frac{\mathbb{P}_{\mathbf{y} \sim \mathcal{D}_{-l}}[\mathbf{y} \in F]}{\mathbb{P}_{\mathbf{y} \sim \mathcal{D}_l}[\mathbf{y} \in F]} \leq \frac{\alpha^{-1} \epsilon_s}{1 - \epsilon_c},$$

which allows us to continue the proof analogously to Lemma B.2. $\qquad \square$

## B.5 Finite and Biased datasets

Real datasets come with further challenges when evaluating the completeness and soundness.

Let us introduce two data distributions $\mathcal{T}$ and $\mathcal{D}$ on the same dataset $D$, where $\mathcal{T}$ is considered the "true" distribution and $\mathcal{D}$ a different, potentially biased, distribution. We define this bias with respect to a specific feature $\phi \in \Sigma$ as

$$d_\phi^l(\mathcal{D}, \mathcal{T}) := |\mathbb{P}_{\mathbf{x} \in \mathcal{D}}[c(\mathbf{x}) = l \mid \mathbf{x} \in \phi] - \mathbb{P}_{\mathbf{x} \in \mathcal{T}}[c(\mathbf{x}) = l \mid \mathbf{x} \in \phi]|.$$

Note that $d_\phi^1(\mathcal{D}, \mathcal{T}) = d_\phi^{-1}(\mathcal{D}, \mathcal{T}) =: d_\phi(\mathcal{D}, \mathcal{T})$ and $0 \le d_\phi(\mathcal{D}, \mathcal{T}) \le 1$. This distance measures if data points containing $\phi$ are distributed differently to the two classes for the two distributions.

For example, consider as $\phi$ the water in the boat images of the PASCAL VOC dataset Lapuschkin et al. (2019). The feature is a strong predictor for the "boat" class in the test data distribution $\mathcal{D}$ but should not be indicative for the real world distribution which includes images of water without boats and vice versa. We now want to prove that a feature selected by $M$ is either an informative feature or is misrepresented in the test dataset.

**Lemma B.5.** *Let* $\mathfrak{D}, k, B, A, M, \alpha, \epsilon_c$ *and* $\epsilon_s$ *be defined as in Theorem 2.12. Let* $\mathcal{T}$ *be the true data distribution on D. Then for* $\delta \in [0, 1]$ *we have*

$$\mathbb{P}_{\mathbf{y} \sim \mathcal{T}}[c(\mathbf{y}) = c(\mathbf{x}) \mid \mathbf{y} \in M(\mathbf{x})] \ge 1 - \delta - d_{M(\mathbf{x})}(\mathcal{D}, \mathcal{T}),$$

*with probability* $1 - \frac{1}{\delta}\left(\frac{k\alpha\epsilon_s}{1 + k\alpha\epsilon_s B^{-1} - \epsilon_c} + \epsilon_c\right)$ *for* $\mathbf{x} \sim \mathcal{D}$.

This follows directly from Lemma 2.7, Theorem 2.12, the definition of $d_\phi(\mathcal{D}, \mathcal{T})$ and the triangle inequality. This means that if an unintuitive feature was selected in the test dataset, we can pinpoint to where the dataset might be biased.

We provided Lemma B.5 in the context of biased datasets. The next iteration considers the fact that we only sample a finite amount of data from the possibly biased test data distribution. This will only give us an approximate idea of the soundness and completeness constants.

**Lemma B.6.** *Let* $D, \sigma, \mathcal{D}, cA, M$ *and* $\mathcal{T}$ *be defined as in Lemma B.5. Let* $D^{test} = (\mathbf{x}_i)_{i=1}^N \sim \mathcal{D}^N$ *be* $N$ *random samples from* $\mathcal{D}$. *Let*

$$\epsilon_c^{test} = \max_{l \in \{-1, 1\}} \frac{1}{N} \sum_{\mathbf{x} \in D_l^{test}} \mathbb{1}(A(M(\mathbf{x}, c(\mathbf{x}))) \ne c(\mathbf{x})),$$

*and*

$$\epsilon_s^{test} = \max_{l \in \{-1, 1\}} \frac{1}{N} \sum_{\mathbf{x} \in D_l^{test}} \mathbb{1}(A(M(\mathbf{x}, -c(\mathbf{x}))) = -c(\mathbf{x})).$$

*Then it holds with probability* $1 - \eta$ *where* $\eta \in [0, 1]$ *that on the true data distribution* $\mathcal{T}$ $A$ *and* $M$ *obey completeness and soundness criteria with*

$$\epsilon_c \le \epsilon_c^{test} + \epsilon_{dist} + \epsilon_{sample} \quad \text{and}$$
$$\epsilon_s \le \epsilon_s^{test} + \epsilon_{dist} + \epsilon_{sample}$$

*respectively, where* $\epsilon_{dist} = \max_{l \in \{-1, 1\}} \|\mathcal{D}_l - \mathcal{T}_l\|_{TV}$ *and* $\epsilon_{sample} = \sqrt{\frac{1}{2N} \log\left(\frac{4}{\eta}\right)}$.

The proof follows trivially from Hoeffding's inequality and the definition of the total variation norm.

*Proof.* We define $E_{c,l} = \{\mathbf{x} \in D_l \mid A(M(\mathbf{x}, c(\mathbf{x}))) \ne c(\mathbf{x})\}$ for $l \in \{-1, 1\}$ and let $E_{c,l}^{\mathcal{D}}$ be the Bernoulli random variable for the event that $X \in E_{c,l}$ where $X \sim \mathcal{D}$. Then

$$\mathbb{P}_{\mathbf{x} \sim \mathcal{D}_l}[A(M(\mathbf{x}, c(\mathbf{x}))) \ne c(\mathbf{x})] = \mathbb{E}[E_{c,l}^{\mathcal{D}}]$$

Using Hoeffding's inequality we can bound for any $t > 0$

$$\mathbb{P}\left[\left|\left(\frac{1}{N} \sum_{\mathbf{x} \in D_l^{test}} \mathbb{1}(\mathbf{x} \in E_{c,l})\right) - \mathbb{E}[E_{c,l}^{\mathcal{D}}]\right| > t\right] \le e^{-2nt^2}.$$

We choose $t$ such that $e^{-2t^2} = \frac{\eta}{4}$. We use a union bound for the maximum over $l \in \{-1, 1\}$ which results in a probability of $2\frac{\eta}{4} = \frac{\eta}{2}$ we have

$$\max_{l \in \{-1, 1\}} \mathbb{E}[E_{c,l}^{\mathcal{D}}] > \epsilon_c^{test} + \sqrt{\frac{1}{2N} \log\left(\frac{4}{\eta}\right)},$$

and thus with $1 - \frac{\eta}{2}$ we have $\max_{l \in \{-1,1\}} \mathbb{E}\left[E_{c,l}^{\mathcal{D}}\right] \leq \epsilon_c^{\text{test}} + \epsilon^{\text{sample}}$. Using the definition of the total variation norm

$$\|\mathcal{T} - \mathcal{D}\|_{TV} = \sup_{J \subset D} |\mathcal{T}(J) - \mathcal{D}(J)|,$$

with $J = E_{c,l}$ we can derive $\mathbb{E}\left[E_{c,l}^{\mathcal{T}}\right] \leq \mathbb{E}\left[E_{c,l}^{\mathcal{T}}\right] + \|\mathcal{T} - \mathcal{D}\|_{TV}$ and thus

$$\epsilon_c \leq \epsilon_c^{\text{test}} + \epsilon^{\text{sample}} + \epsilon^{\text{dist}}.$$

We can treat $\epsilon_s$ analogously and take a union bound over both the completeness and soundness bounds holding which results in the probability of $1 - \eta$. □

## C  NUMERICAL EXPERIMENTS

For the numerical experiments, we implement Arthur, Merlin and Morgana in Python 3.8 using PyTorch (BSD-licensed). We perform our experiments on the UCI Census Income dataset and the MNIST dataset, which is licensed under the Creative *Commons Attribution-Share Alike 3.0* licence.

### C.1  TRAINING SETUP FOR CENSUS INCOME DATASET

In the following, we provide an overview of the experiments performed on the Census Income dataset, including preprocessing steps and training configurations.

**Data Preprocessing**  The UCI Census Income dataset consists of both continuous and categorical features, 14 features in total. The target class is chosen to be the feature "sex", which contains the categories "male" and "female", to indicate a possible case of discrimination. The feature "fnlwgt" is removed from the set of features, since it does not contain any meaningful information. In addition, the features "marital-status" and "relationship" are also removed as they strongly indicate the target class. After removal, 11 features remain for each data point. The continuous features are scaled according to the min-max scaling method. To simplify feature masking, all features are padded to a vector of the same fixed dimension. Continuous features are then repeated along the padding dimension, while categorical features are one-hot encoded to the length of the fixed dimension. Note, that the fixed dimension is determined by the categorical feature with the most categories.

Finally, the train and test datasets are balanced with respect to the target class, resulting in 19564 train samples and 9826 test samples.

**Model Description**  Arthur is modelled as a NN with a single hidden linear layer of size 50 followed by a ReLU activation function. The output is converted to a probability distribution via the softmax function with three output dimensions, where the third dimension corresponds to the "Don't know!" option. The resulting NN contains approximately 23k parameters. Merlin and Morgana, on the other hand, are modelled as Frank-Wolfe optimisers, which are discussed in more detail in the overview of the experiments conducted on the MNIST dataset.

| Model | Learning Rate | Batch Size | Frank-Wolfe Learning Rate | Epochs |
|---|---|---|---|---|
| Pre-training Arthur | $10^{-4}$ | 512 | - | 100 |
| Merlin-Arthur Classifier | 0.5 | 512 | 0.1 | 20 |

Table 1: Configurations for pre-training Arthur and training the Merlin-Arthur classifiers.

**Model Selection**  The training and testing of the models is divided into two phases. First, we pre-train Arthur on the preprocessed UCI Census Income dataset without Merlin and Morgana. Second, we use the pre-trained model to perform further training including Merlin and Morgana. For the pre-training of Arthur, our experiments were conducted such that the models' parameters were saved separately after each epoch. Consequently, for each experiment, we had access to a set of candidate models to choose from for further analysis. To ensure high completeness, we select the pre-trained candidate with the highest accuracy with respect to the test dataset. The selected

model then represents the pre-trained classifier for all subsequent experiments involving Merlin and Morgana. Similar to the pre-training process, we also stored the model parameters after each epoch of training with Merlin and Morgana. The results presented in Figure 5 correspond to the models with the highest completeness at a soundness of at least 0.9 among all epochs. The training configurations regarding the pre-training of Arthur and the training of the Merlin-Arthur classifiers can be obtained from Table 1.

---

**Algorithm 2** Merlin-Arthur Training

---

1: **Input:** dataset: $D_{\text{train}}$, Epochs: $N, \gamma$
2: **Output:** Classifier Network (A), Optional: Masking Networks Merlin $(M)$ and Morgana $(\widehat{M})$
3: **for** $i \in [N]$ **do**
4:     **for** $\mathbf{x}_j, \mathbf{y}_j \in D_{\text{train}}$ **do**
5:         $\mathbf{s}_M \leftarrow M(\mathbf{x}_j, \mathbf{y}_j), \mathbf{s}_{\widehat{M}} \leftarrow \widehat{M}(\mathbf{x}_j, \mathbf{y}_j)$      $\triangleright$ $M$ and $\widehat{M}$ can be the optimiser or the NN
6:         $A \leftarrow \arg\min_A (1 - \gamma) L_M(A(s_M \cdot x_j), y_j) + \gamma L_{\widehat{M}}(A(s_{\widehat{M}} \cdot x_j), y_j) \triangleright$ Update classifier
    using masked images
7:         $M \leftarrow \arg\min L_M(A(M(\mathbf{x}_j) \cdot x_j), y_j)$                $\triangleright$ Update only if $M$ is a NN
8:         $\widehat{M} \leftarrow \arg\max L_{\widehat{M}}(A(\widehat{M}(\mathbf{x}_j) \cdot x_j), y_j)$            $\triangleright$ Update only if $\widehat{M}$ is a NN
9:     **end for**
10:    **for** $\mathbf{x}_j, \mathbf{y}_j \in D_{\text{train}}$ **do**
11:       $A \leftarrow \arg\min_A L(A(x_j), y_j))$              $\triangleright$ Update classifier using regular images
12:    **end for**
13: **end for**

---

## C.2   TRAINING SETUP FOR MNIST

Here, we give a detailed description of our training setup for MNIST and show the error bars of the numerical results presented in the main part of the paper over 10 different training runs.

*Structure of Arthur.* Arthur is modelled using a neural network. Specifically, we consider a convolutional NN with a ReLU activation function. For the case of 2 classes, we consider a NN with 2 convolution layers, whereas for the 5 class case we consider 3 convolution layers. The output of the convolution is then passed through 2 linear layers before being used for the output. Table 2 describes the used architecture in detail.

| Layer Name | Parameters |
|---|---|
| Conv2D | I=3, O=32, K=3 |
| ReLU | |
| Conv2D | I=32, O=64, K=3 |
| ReLU | |
| Conv2D | I=64, O=64, K=3 |
| ReLU | |
| MaxPool2d | K=2 |
| Linear | I=7744, O=1024 |
| ReLU | |
| Linear | I=1024, O=128 |
| ReLU | |
| Linear | I=128, O=1 |

Table 2: Description of the neural network architecture used for the Arthur classifier.

*Structure of Merlin and Morgana.* Recall that Merlin and Morgana aim to ideally solve

$$M(\mathbf{x}) = \arg\min_{\mathbf{s} \in B_k^d} L_M(A, \mathbf{x}, \mathbf{s}) \quad \text{and} \quad \widehat{M}(\mathbf{x}) = \arg\max_{\mathbf{s} \in B_k^d} L_{\widehat{M}}(A, \mathbf{x}, \mathbf{s}),$$

respectively, where $L_M$ and $L_{\widehat{M}}$ are the loss functions defined in Section 3, and $B_k^d$ is the space of $k$-sparse binary vectors. Thus, Merlin and Morgana take an image as input and produce a mask

of the same dimension with $k$ one-entries and zero-entries otherwise. We additionally added a regularisation term in the form of $\lambda\|\mathbf{s}\|_1$ to both of the objectives, and set $\lambda = 0.25$. We realise the pair Merlin and Morgana in four different ways, which we explain in the following. All of these approaches return a mask $\mathbf{s} \in [0,1]^d$ that we then binarise by setting the $k$ largest values to one and the rest to zero.

**Frank-Wolfe Optimisers**  In the first case, Merlin and Morgana are modelled by an optimiser using the Frank-Wolfe algorithm (Jaggi, 2013). We follow the approach outlined by Macdonald et al. (2021) with the Frank-Wolfe package provided by Pokutta et al. (2020). The optimiser searches over a convex relaxation of $B_k^d$, i.e.,

$$\mathcal{B}_k^d = \left\{ \mathbf{v} \in [0,1]^d \,\middle|\, \|\mathbf{v}\|_1 \le k \right\},$$

the $k$-sparse polytope with radius 1 limited to the positive octant. To optimise the objective we use the solver made available at `https://github.com/ZIB-IOL/StochasticFrankWolfe` with 200 iterations.

**UNet Approach**  For the second case, we model Merlin and Morgana using NNs, specifically a UNet that has already been used in the XAI domain to create saliency maps, see (Dabkowski & Gal, 2017). We copy the UNet design by Ronneberger et al. (2015) since it achieves good results in image segmentation, a task reasonably similar to ours. We predict mask values between zero and one for each image pixel, and rescale the mask should it lie outside of $\mathcal{B}_k^d$. The binarisation of the mask is ignored during the backpropagation that trains the UNets and only employed to produce the masks that Arthur is trained on.

**Hybrid Approach**  In the Hybrid approach, Merlin is modelled by a UNet, whereas Morgana is still modelled by the FW-optimiser. This approach is useful since for a sound Arthur that cannot be fooled, the training of the Morgana UNet becomes difficult and the UNet diverges. It then takes a while of training for the UNet to adapt, should Arthur open himself to possible adversarial masks. The optimiser is applied to each individual image instead and can find possible weaknesses instantly.

**Class-Networks**  One of the alternatives to Merlin and Morgana that we propose are class-specific UNets. Instead of Merlin and Morgana each being represented by a network, there is a UNet associated with each class that is trained to produce a feature mask that convinces Arthur of its own class for any input image, i.e., for $l \in [C]$ try to solve

$$M_l(\mathbf{x}) = \underset{\mathbf{s} \in B_k^d}{\arg\min} -\log(A(\mathbf{s} \cdot \mathbf{x})_l) + \lambda\|\mathbf{s}\|_1.$$

Merlin is then implemented as an algorithm to choose the UNet corresponding to the true class, so

$$M(\mathbf{x}) = M_{c(\mathbf{x})}(\mathbf{x}).$$

Morgana instead uses for each individual image the output of the UNet that most convinces Arthur of a wrong class (maximises the Morgana-loss), i.e.,

$$M(\mathbf{x}) = M_l(\mathbf{x}) \quad \text{with} \quad l = \underset{l \in C \setminus \{c(\mathbf{x})\}}{\arg\max} L_{\widehat{M}}(A, \mathbf{x}, M_l(\mathbf{x})).$$

Ideally, this training setup would be more stable than the normal UNet approach. When Arthur cannot be fooled, the class-UNets still have a reasonable objective in convincing him of the correct class, which hopefully prevents divergence as for the Morgana UNet. Experimentally, however, the class-networks proved to be much less stable than the simple UNet approach, see Figure 15.

We give an overview over the parameters used for the four different approaches in Table 3.

**Merlin-Arthur Classifier Training**  The overall training procedure proceeds as outlined in Algorithm 2. We initially train Arthur directly on the training data. In the optimiser approach, this pre-trained network is used to search for the optimal masks for Merlin and Morgana. In the UNet approach, these masks are directly produced by the UNets. Arthur is then trained on the masked images over the whole dataset. The UNets are then trained on the dataset with a fixed Arthur according to their respective objectives. We cycle through this process for 5 epochs. The learning rate is 1e-4 for the Arthur and Merlin network and 1e-6 for the Morgana and the class-specific networks.

| Parameter | Value |
|---|---|
| Batch Size | 128 |
| Baseline Value | 0.3 |
| Max FW Iterations | 200 |
| FW Momentum | 0.9 |
| Regularisation $\lambda$ | 0.25 |
| Max NN Passes | 5 |
| Arthur LR | 1e-4 |
| Merlin LR | 1e-4 |
| Morgana LR | 1e-6 |

Table 3: Values of various parameters used in training.

## C.3 PURELY COOPERATIVE SETUP AND "CHEATING" FOR MNIST

Here, we discuss the Merlin-Arthur classifier when only Merlin is used with no Morgana. Our results demonstrate that the inclusion of Morgana is necessary for Merlin and Arthur to exchange meaningful features and abstain from "cheating". For a purely cooperative setup, information about the class $c(\mathbf{x})$ that Arthur infers is dominated by the fact that Merlin chose that feature, rather than the feature itself, i.e., $H(c(\mathbf{x})|M(\mathbf{x}) = \phi) \ll H(c(\mathbf{y})|\mathbf{y} \in \phi)$. We can upper bound $H(c(\mathbf{x})|M(\mathbf{x}) = \phi)$ through the classification error $P_e$ that Arthur and Merlin achieve via Fano's inequality (Fano, 1961):

$$H(c(\mathbf{x})|M(\mathbf{x}) = \phi) \leq H_b(P_e) + P_e \log(|C| - 1),$$

where $|C|$ is the number of classes. We can then bound the amount of information that is transferred by the choice of the mask rather than the feature itself as

$$I(c(\mathbf{x}); M(\mathbf{x}) = \phi) - I(c(\mathbf{y}); \mathbf{y} \in \phi) = H(c(\mathbf{y}) \mid \mathbf{y} \in \phi) - H(c(\mathbf{x}) \mid M(\mathbf{x}) = \phi)$$
$$\geq H(c(\mathbf{y}) \mid \mathbf{y} \in \phi) - (H_b(P_e) + P_e \log(|C| - 1)).$$

We define the *cooperative information* as

$$I_{\text{coop}} := \max\left(0, H(c(\mathbf{y}) \mid \mathbf{y} \in \phi) - (H_b(P_e) + P_e \log(|C| - 1))\right),$$

which lower bounds how much Arthur and Merlin "cheat". We train Merlin and Arthur on the MNIST dataset and show the results in Figure 14 that in the purely cooperative case $I_{\text{coop}}$ keeps the classification error low despite exchanging uninformative features. Including Morgana on the other hand pushes $I_{\text{coop}}$ to zero even for small masks. In this case Merlin produces highly informative features.

Figure 14 depicts "Cheating" measured by $I_{coop}$ over the MNIST test dataset (restricted to 5 classes) for different setups. In a purely cooperative setup, Arthur and Merlin learn to communicate over uninformative features. Thus, the classification error $P_e$ stays low, while the conditional entropy of the class with respect to the feature goes up. When Morgana is present and soundness is high (in our analysis $\epsilon_s < 0.05$), Arthur and Merlin cannot cheat, the classification error increases for small $k$.

Why is this observation important? Note that the purely cooperative setup could already be seen as interpretable. With Merlin as a network it appears as a version of a self-interpreting network. With Merlin as an optimiser, it is similar to Rate-Distortion Explanations (RDE) (Macdonald et al., 2019). In fact, RDE have been criticised for producing "cheating" masks that artificially create features that were not present in the original image (Wäldchen et al., 2022). We connect this to the lack of robustness with respect to Morgana. Designers of self-interpreting setups need to explain how they circumvent the problem of "cheating" masks.

Figure 15 depicts the results from the main paper in more detail. Specifically, it depicts averages and the standard deviation as error bars for each parameter using samples from 10 different training runs. The results presented here are consistent with those presented in the main body of the paper, except for the class networks. The error bars are large for small mask sizes, but shrink as the mask size increases. The class-network approach is considerably less stable than our other implementations, even though it was conceived as a stabler alternative to the simple network approach. One possibility might be that since each UNet both cooperates with Arthur and wants to fool him, they are more

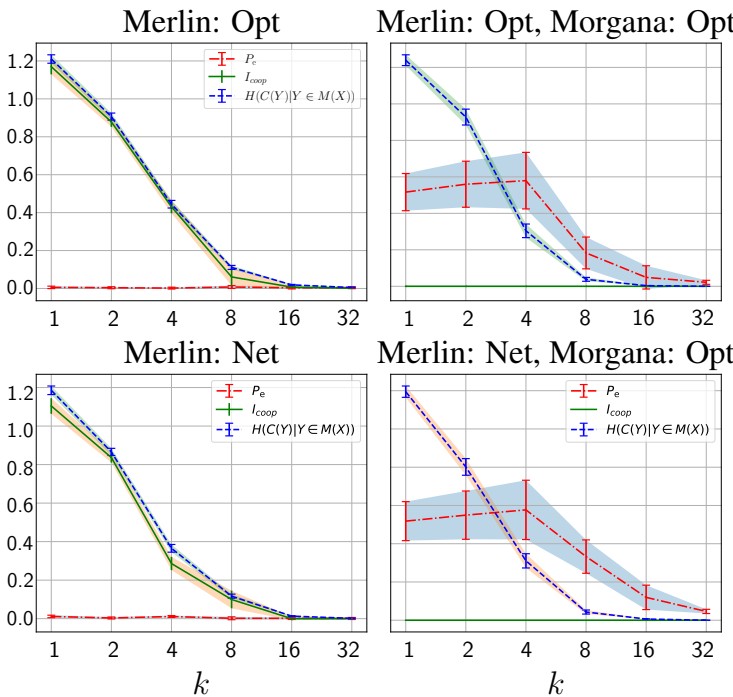

Figure 14: This figure depicts the mean and standard deviation over 10 training runs for the error probability, cooperative information and the class entropy. This was obtained from 5-class classification with classes 1,2,3,4 and 5 with $\gamma = 0.75$ for the purely cooperative setup (*left*) and the adversarial setup (*right*), where Merlin was realised as an optimiser (*top*) or as a neural network (*bottom*).

sensitive when Arthur changes his focus between achieving good soundness and completeness. We hope that further research will determine if this realisation can be trained in a stable manner.

We also conduct experiments to illustrate the impact of varying values of $\gamma$. The results of these are depicted in Figure 16. As expected, increasing the value of $\gamma$ increases the error rate as it gives more weight to Morgana. This increase is not visible in the NN only setup, indicating this setup may not be creating high quality masks.

## C.4 CENSUS INCOME EXPERIMENTS WITH MARITAL-STATUS AND RELATIONSHIP INCLUDED

In Section 3.1, we demonstrate how an adversarial setup prevents a typical approach to manipulating explanations that has been demonstrated for numerous XAI methods. To show the effect most clearly, we kept the focus on a single sensitive feature ("sex") and removed the "relationship" and "marital status" features that strongly correlate with it and are thus informative. We now show the results for the experiments including these features in Figure 17, where Merlin uses these features to communicate "sex" with a completeness of 71%.

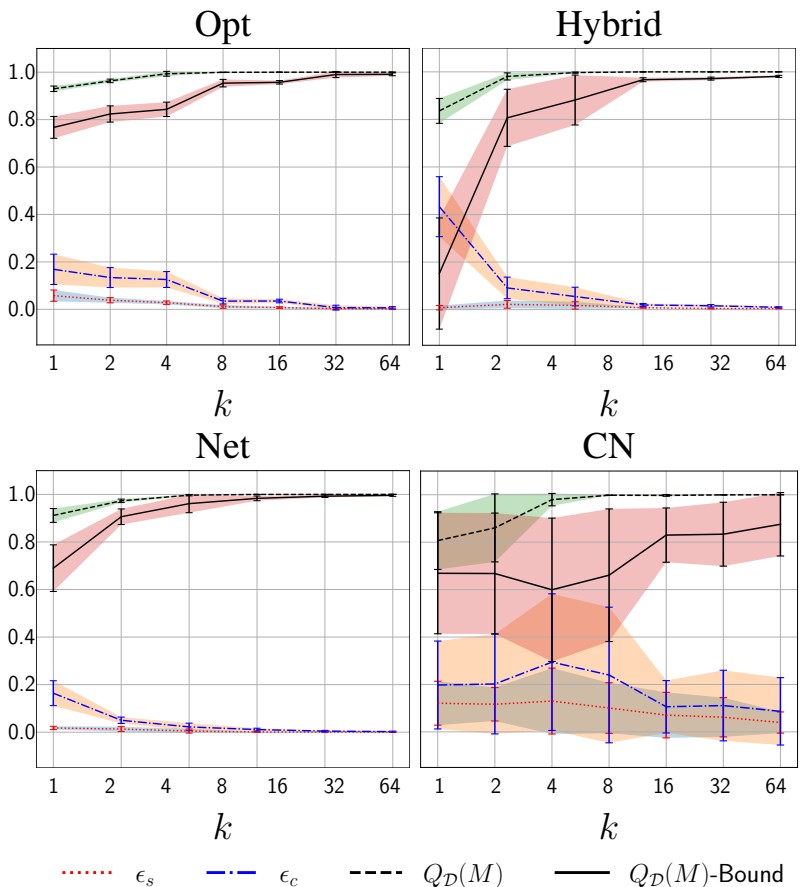

Figure 15: We show the mean and standard deviation over 10 training runs for completeness and soundness, along with the average precision and its bound as obtained from 2-class classification with classes 6 and 9 with $\gamma = 0.75$ for different settings.

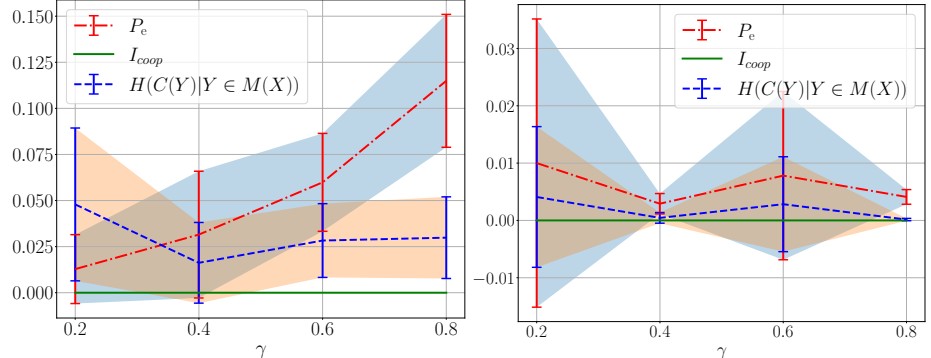

Figure 16: This figure depicts the error probability, cooperative information and the class entropy as obtained from 2-class classification with classes 6 and 9 for different values of $\gamma$ at a mask size of 16 for the Hybrid (left) and Network (right) settings

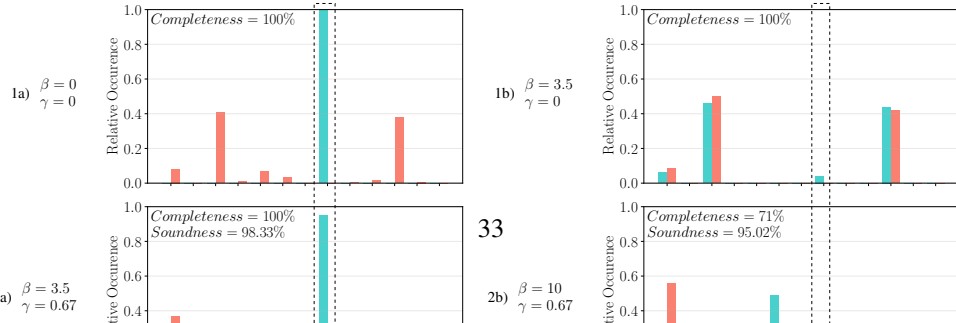

