# OpenReview forum: "Formal Interpretability with Merlin-Arthur Classifiers"
_ICLR.cc/2023/Conference — Submitted to ICLR 2023_

### Official Review · Reviewer_LYB7 · 2022-10-20

**Confidence:** 3
**Correctness:** 2
**Technical Novelty And Significance:** 3
**Empirical Novelty And Significance:** 3
**Recommendation:** 5

**Clarity, Quality, Novelty And Reproducibility:**

As discussed above, clarity is a serious issue with this work, and some of the claims have little to no evidence provided. This isn't an issue with reproducibility exactly, but it does impact the quality of the work. It seems very preliminary.

On the other hand, there is an idea here that may have potential, and an interesting perspective. So I'd give high marks for novelty overall.

**Strength And Weaknesses:**

The main strength of this paper is that it does seem to suggest a way to obtain reasonable explanations while still using modern neural network methods. The adversarial training method overcomes some kinds of spurious "explanations" that may crop up when merely enforcing, e.g., sparsity of the chosen features.

This paper may have potential, but the current submission has some serious weaknesses:

- The key quantities of "AFC" and imbalance of the prover/adversary that underlie the theoretical guarantee are named and even suggested to be helpful in thinking about interpretability, but they are *not* defined or sketched in the paper. I know they are in an appendix, but I'm not sympathetic to attempts to circumvent the page limit by punting the main concepts to the appendix. It doesn't bode well for their conceptual clarity when they could not be described in the main submission. It also makes it difficult to judge the strength of the theorems. I have no idea whether or not the assumption that these quantities are small constants is actually reasonable, and no evidence is presented for the assumption.

- The experiments are very limited: there is one data set for each point the paper seeks to make regarding manipulation and effectiveness, respectively the UCI census data set and MNIST. (Incidentally, Deng 2012 is certainly not the origin of MNIST... it's LeCun et al. 1998) The experiments are thus illustrative but not so convincing: how broadly do the claims hold?

- While explanations using a small number of features is generally desirable, the method as presented seems to scale exponentially with the number of features, which may not be adequate for more complex tasks -- for example, even for a task like CIFAR, it isn't clear to me how effective this would be. (This also ties into the previous issue)

- (relatively minor) The formalism of feature spaces used in the paper seems to only serve to obfuscate the presentation. As far as I can tell, it plays no actual role in the development, and ultimately the explanations are just fixing small subsets of the coordinates to given values when viewing each input as a vector. (If the authors are looking for something to cut in order to introduce the key quantities, this discussion does more harm than good in my opinion.)


**Summary Of The Paper:**

This paper proposes to train classifiers with guaranteed Anchors-style interpretability by structuring the classifier in two parts: a feature selector and a classifier that is only provided the selected features. During training, the classifier portion is also subject to adversarial feature selectors. The features selected by the first component thus may serve as an explanation of the classifier's output: under some conditions on the data and families of classifiers, the chosen features are sufficient to indicate the class. Some simple experiments suggest that the method can be applied to neural networks that achieve good performance in practice (on MNIST...) and illustrate resistance to manipulation.

Incidentally, although the paper suggests that it is inspired by "interactive proofs," there is not much interaction to speak of: the prover sends a message to a verifier who accepts or rejects. It's much more accurate to say that it's an NP style certificate. Calling it an interactive proof only makes it misleading to those who know what that means, and mysterious to those who don't.

**Summary Of The Review:**

I find the idea interesting and I hope the authors give it the presentation and evaluation it deserves.

---

> ### Author Response · Authors · 2022-11-10
> **Response to "Weaknesses"**
>
> We thank the reviewer for their interest and detailed questions and are happy to address their concerns.
>
> 1. We will bring forward the definition of AFC and relative success rate to the main document. Please see our general answer for why we think that the AFC assumption is reasonable.
>  Note that our assumption about the AFC being 1 is only used for numerical evaluation of the bound on average precision and it is not needed for the theoretical results.
>  2. Our theoretical results do not make any significant assumptions about the data and thus hold very broadly. Our numerics are comparable in scope to [1] and [2] of reviewer 2.  Experiments for more complicated data sets will be explored in further research. We thank the reviewer for the correction of our citation and will include this change in the paper.
> 3. If implemented as in our experimental section, the difficulty does not scale exponentially. While guarantees for feature quality usually would require an exponentially costly exhaustive search (such as Proposition 1 in ref [5] of reviewer 3), for our bound in Theorem 2.10. this is not the case. We only need a comparable strength for Merlin's and Morgana's algorithm. Thus, our approach reduces this problem to the training of multiple neural networks which is significantly easier. While increasing the feature size will make optimization more difficult, the existence and use of large scale generative adversarial networks indicates that this is possible.
> 4. We thank the reviewer for this suggestion and we will simplify our definition.

---

### Official Review · Reviewer_zBZW · 2022-10-25

**Confidence:** 3
**Correctness:** 2
**Technical Novelty And Significance:** 3
**Empirical Novelty And Significance:** 3
**Recommendation:** 5

**Clarity, Quality, Novelty And Reproducibility:**

* A few elements of the technical presentation are not clear (see above).
* If my understanding is correct, the approach is very interesting and novel.

**Strength And Weaknesses:**

The framework is very interesting, novel and valuable to the community. At the same time, several components of the presentation are unclear, and I parts are severely overstated, ignore previous results, and should be addressed before publication.


1. Respectfully, I find a lot of the technical presentation and definitions confusing and, at best, non standard. Let me elaborate:

a) A "dataset", D, is never defined. The word "dataset" is typically used to refer to a finite collection of samples drawn from a distribution (see [1] and [2] for classic references). It seems that this might not the case here, however, because (looking at Example 2.2, $D\subset [0,1]^d$, so $D$ can be interpreted rather as a sample space. On the other hand, in other parts of the manuscript, the authors insist of having $D$ be a collection of data. This leads to (at least) two other problems: i) is this collection finite? It seems that it is, otherwise the second loop in Algorithm 1 would never finish, and ii) If it is finite, and the models are trained on this, all of the analysis regards the features that are obtained from this finite training set, and not on the expected performance of the respective metrics over a new sample.

b) The authors choose to define "feature" in terms of *subsets of possible samples* from a dataset. This is not what we refer to features in statistical learning (again, see [1,2] for definitions). This leads to other confusing (if correct?) statements, like: "$x$ **contains** a feature $\phi$ if $x \in \phi$". The notation also makes it unclear to understand how features depend on specific samples. E.g., in Eq. (2), $M^*$ depends on a sample $x$, but it's not clear to the reader that $\phi$ is a subset of a specific sample $x$.

2. A few aspects of the presentation of this work are highly overstated: the authors insist this is "the first provable interpretability guarantees". This is incorrect for a couple of reasons:

a) There is a number of approaches that provide probably correct explanations for machine learning classifiers. All of these rely on certain assumptions, that vary in complexity. Moreover, in many settings these algorithms are not only correct but also efficient. E.g., the work [3] presents a linear time algorithm for instance wise feature importance scoring on black-box models based on the Shapley framework. The work in [4] does something similar, providing linear time algorithms for provably interpretability of imaging data for general predictors under mild assumptions.

b) Importantly, the methods above (as the majority of the explainability methods for 'off-the-shelf' predictors) are applicable to any given classifier. The method proposed in this work *is not*, and cannot be applied to any given model. Instead, this manuscript proposes to train *a new* classifier (Arthur) that is inherently interpretable, as it is so that it relying only on specific features. This is a totally valid strategy, and indeed a very interesting alternative approach to interpretable ML, but this difference should be made clear and precise to the reader. Moreover, work on this vein has also been explored in what I believe is the most closely related work to this manuscript [5]. Similar to this work, the work in [5] also provides provably correct interpretable predictors that rely on a finite number of features (or queries).

c) Even while the authors provide guarantees for their general method, the application of these ideas do not inherit these nice properties because -as the authors state- convergence to an equilibrium point is not guaranteed, and since their models are trained on finite datasets only. Furthermore (probably inheriting my confusion from point (1)), the results seem to hold only for the training set.

Given all these points, the statement that the authors provide "the first provable interpretability guarantees", and its variations that appear throughout the text, are grossly incorrect. The authors should moderate these comments, and they should better situate their contribution in light of these previous results. The authors should not worry about this, however - their proposed method is very interesting regardless!


3. An important limitation of the presented approach, that is stated simply in passing, is that they can only consider realizable learning problems: they consider that the label is determined by a concept function c(x) (i.e. the Bayes Risk is 0). The authors should consider making this very explicit, as it is far from a mild assumption in a statistical learning problem (e.g. this likely does not hold for the UCI Census Income dataset).

4. After presenting their main result, and commenting on the implications of the theorem 2.8, the authors informally introduce the notion of "asymmetric feature concentration (AFC)", $\kappa$. From their explanation, it's clear that this is a key property of classification problems, and Lemma 2.9 attempts to characterize this (providing an upper bound to $kappa$). However, $k$ is never defined formally, so it's basically impossible to interpret the utility of Lemma 2.9. This AFC is later formalized and defined in the appendix, but is important to the reader to have some aspects of this more precise definition in the main corpus of the story - otherwise, there's little utility in having Lemma 2.9 in the main body of the paper.

5. Towards the end of section 2, the authors comment on "biased samples", but they never define or otherwise make precise what they mean. Could you clarify?

6. Lastly, the empirical validation is (imho) a bit limited. Moreover, the authors comment that for the Census Income dataset, they remove the 'relationship' and 'marital status' features as they were strongly correlated with the variable 'sex'. Does this mean that the method is very sensitive to correlated features? Why would this be the case? I suggest the authors to include these results in their paper too, and comment on this limitation. This can only clarify the presentation of your method.


[1] Shalev-Shwartz, Shai, and Shai Ben-David. Understanding machine learning: From theory to algorithms. Cambridge university press, 2014.
[2] Mohri, Mehryar, Afshin Rostamizadeh, and Ameet Talwalkar. Foundations of machine learning. MIT press, 2018.
[3] Chen et al. "L-shapley and c-shapley: Efficient model interpretation for structured data." ICLR 2019.
[4] Teneggi. "Fast hierarchical games for image explanations" IEEE Transactions on Pattern Analysis and Machine Intelligence, 2022
[5] Chattopadhyay, Aditya, et al. "Interpretable by Design: Learning Predictors by Composing Interpretable Queries." arXiv preprint arXiv:2207.00938 (2022).

**Summary Of The Paper:**

This paper proposes a new framework for interpretable machine learning relying on interactive proofs ideas. The approach relying the adversarial training of two agents resulting in the most informative features for a given task. The authors provide theoretical results on the performance of the agents, as well as demonstration on a few numerical settings.

**Summary Of The Review:**

In summary, my current understanding of the method (given the confusing definitions in point (1)) leads me to believe that the approach described here is very interesting and a nice contribution. To be certain about this, however, I would need clarification on my comments above, and potentially re defining some of these elements and statements. In closing, I want to stress that while some of my comments above might appear as requiring the authors to make the limitations of their method more explicit and clearer, it is the opinion of this reviewer that this will only increase the value of this contribution - and with it, my support for this paper.

---

> ### Author Response · Authors · 2022-11-09
> **Response to "Weaknesses"**
>
> We thank the reviewer for their interest and detailed questions and are happy to answer.
>
> 1. The dataset $D$ in our formulation is an abstract, possibly infinite set, e.g., the set of images of handwritten digits. $\mathcal{D}$ is then a distribution on this dataset. The training dataset (e.g. MNIST) would be a finite sample from this distribution. We will thus change the $D$ in Algorithm 1 to $D_\text{train}$.
> The theoretical analysis uses expectation values over the whole distribution $\mathcal{D}$.
> For our numerical investigation, we use the test dataset as a faithful sample from this distribution.
> We refer to Appendix B.5 for a detailed description for this approximation.
> 2. Please see our comments on notation in the general answer.
> 3. The cited works indeed approximate Shapley values efficiently. However, even when computed exactly, these Shapley do not guarantee interpretability with respect to the true data distribution, see Appendix A2. Their values are computed by blacking out parts of the images as new inputs for the classifier. These new inputs do not reflect the true distribution of images anymore. There have been attempts to rectify this by modelling the data distribution more exactly, as in Frye et al. and Aas et al. However, these references do not provide quantitative bounds. Please note that while Merlin and Arthur exchange images that are not on-distribution, our mutual information bounds are with respect to the true data distribution. To draw the connection between those two distributions is why the adversary is necessary in the first place.
> 4. We thank the reviewer for this reference that also aims to efficiently find features with high mutual information. We will highlight this work in our related work section.
> However, we believe that this work does not yet provide concrete bounds for feature quality.
> This publication solves the problem of calculating the mutual information practically by training a network to model the joint distribution of queries and datapoints. For interpretability bounds similar to ours, the authors would need some quality measure for how well their network approximates the true data distribution.
> 5. Theorem 2.10 does not rely on an equilibrium, it only considers soundness and completeness of the observed strategies to bound the information content of the exchanged features. There could well be better strategies with better features, but this is not relevant for the bound.
> Please see Appendix B5 for a treatment of the finite sample case. As long as we can assume that the test dataset was faithfully sampled from the true data distribution, the estimates of completeness and soundness on the test set will be close to exact and thus also the bound on the precision. There is no limitation to the training dataset.
> 6. This is correct and we thank the reviewer for highlighting this fact. We will stress this assumption more clearly in the main paper. Extending the results to treat the more general case is left for further research.
> 7. Yes, we agree and will move the definition of the AFC to the main part of the paper. `Biased samples' means that there can be a sample bias between the test dataset and the true underlying data distribution, we refer to Appendix B.5 for a definition.
>         A sampled dataset might only show images of water with boats, whereas the true distribution contains images of water without boats.
>         If the water feature is exchanged even though it is not indicative with respect to the true distribution, it highlights the bias of the sampled dataset. This has been observed, see Lapushkin et al.
> 8. Our numerical evaluation is a proof of concept of comparable difficulty to the datasets of the work that was referenced by the reviewers.
>         We removed 'marital status' and 'relationship' to make the effect of the cheating prevention clearer to the reader. If we include them, then Merlin can communicate 'sex' up to an accuracy of 71\% via these features. But this does not constitute cheating in our sense, since these features do indeed have high mutual information with 'sex'.
>         For an application in discrimination prevention, these informative features then have to be further investigated.
>         But this issue is shared by all feature-based interpretability approaches and goes far beyond the scope of this paper. We agree to include the results with these features in the appendix.

---

> > ### Comment · Reviewer_zBZW · 2022-11-28
> > **Thank you for the clarifications**
> >
> > Dear authors,
> >
> > Thank you for your clarifications (and apologies for the latency as I tended to personal matters).
> >
> > I appreciate the authors responses, which have clarify most of my doubts. A couple of follow up comments:
> >
> > i. It is still my understanding that their approach only address realizable settings - can you please confirm if this is the case, or whether the presented approach could be relaxed to consider more general settings? In either case, this should be clearly stated in the paper.
> >
> > ii. I much appreciate the comprehensive discussion on related works, which now position this contribution better. This reviewers believes, nonetheless, that their comments regarding Shapley coefficients is not correct. It's true that some works have done this by simply obfuscating certain pixels/features, and this is incorrect (as the authors point out). Yet, once sampling is done correctly (by sampling from the respective conditional distribution of the data, as noted by Frye et al, Aas et al, and others), these coefficients do in fact convey precise notions of interpretability (see Teneggi et al. "From Shapley back to Pearson: Hypothesis Testing via the Shapley Value").
> >
> > iii. I also see that the authors chose to preserve their non-standard definitions and notation (despite three reviewers raising questions about this). This is of course understandable, alas comes at the expense of clarity of presentation.

---

> > > ### Author Response · Authors · 2022-11-29
> > > **Further Response**
> > >
> > > Dear reviewer,
> > >
> > > Thank you for your time writing a response and further interest in the paper, we are happy to provide answers.
> > >
> > > **i.** We do not make statements about learnability, which is why we did not discuss realisability in the paper.
> > > Our theorems are not based on any assumptions on realisability, thus our statements hold even in the non-realisable case (which is the more realistic one). We see two ways in which realisability could be considered. First, whether the target $c(x)$ belongs to the set of functions that can be realised by the Merlin-Arthur classifier. Second, whether the optimal strategies for Merlin, Arthur and Morgana can be realised by their respective algorithms.
> > >
> > > To the first point: our statements are not concerned with generalisation. Soundness and completeness are always determined on the test set. It might well be that for a fixed hypothesis class of Arthur and Merlin the function $c(x)$ is not learnable, or that they overfit on the training data. However, both would be reflected in the test set measurements. Practically, neural networks learn well on a variety of datasets and we do not expect generalisation to be an issue even for more complicated data.
> > >
> > > Regarding the second point: in our Theorem 2.12., we do not rely on optimal algorithms, but rather on the already relaxed notion of relative success probability between Morgana and Merlin. Crucially, our class of strategies is thus much more general than the one considered by Chang et al. For their theoretical results, the authors restrict the provers to select single features individually without context. This requirement is later dropped for the numerical experiments.
> > > While our theoretical bound holds independent of realisability, the bound is only useful if $\alpha \gg 0$, i.e., if a strategy with comparable success rate is available for Morgana to compete with Merlin. We give an artificial example of a dataset where this is not the case for poly-time provers in Figure 13 in Appendix B4. Thus, our Assumption 2.13. corresponds to the assumption that such effects will not appear in real-world data.
> > >
> > > We hope this addresses the question. If not, further description of realisability would help us to answer you better.
> > >
> > > **ii.** We are in agreement here. If one does indeed sample from the conditional distribution of the data, then these Shapley values can be efficiently approximated and are very useful for interpretability. But this conditional distribution has to be modelled. On the practical side, there are a number of approaches that achieve this (including Aas et al., Frye et al.). But to derive theoretical bounds on the Shapley values, one must first have bounds on the quality of the model with respect to the ground truth distribution. This is very difficult to achieve. Even if one would have such a bound, e.g., in terms of the Kullback-Leibler divergence between the model and a large test dataset, this bound does not necessarily hold for a conditioned version of the distribution. So one would need a bound for every reasonable conditioning. At least to our knowledge, there are no approaches that achieve that for complex data.
> > >
> > > Teneggi et al. connect the Shapley values to independence of the features and the class. However, they assume that the true conditional distributions are known. In their experiments they use either synthetic data, where they are known explicitly, or medical data, where the pixels out of coalition are modelled by a neutral background image, see Figure E.1. on page 22.
> > >
> > > **iii.** Thank you for your understanding. We had considered a variety of options to formulate the theory in a different notation. However, we either would have to introduce new symbols, which are then again hard to parse. Or would need to exclude the application to more abstract feature spaces, which is important to our argument in Appendix A.4 and further work on this topic considering prover-verifier games on explicit feature spaces modelled as graphs.

---

### Official Review · Reviewer_Pt1p · 2022-10-26

**Confidence:** 4
**Correctness:** 4
**Technical Novelty And Significance:** 2
**Empirical Novelty And Significance:** 2
**Recommendation:** 5

**Clarity, Quality, Novelty And Reproducibility:**

**Clarity and quality:** The paper is written very well and clearly. The theorems/proofs are also written clearly.

**Novelty:** As stated above, there exist prior work that has a lot of overlap with the claimed contributions of the submission. The CAR framework of [1] and the prover-verifier framework of [2], along with the mutual information bounds of [1] capture an important portion of the claimed novelty. These works are not cited by the submission. The bounds that use the concept of AFC do appear to be novel (and interesting).

**Reproducibility:** Considering the code release and the details presented in the appendix, the results in the submission are reproducible.


**Typos:** The paper is very well written and has very few typos -- just listing a few that I've come across in case it helps the authors.
* Section 2.3, First paragraph: The Fyre citation should be in citep.

**Strength And Weaknesses:**

**STRENTGHS:**
* **Important and timely research direction:** Interpretability methods that provide guarantees is a very important research direction, and the discussed method tackles this very important problem.
* **Clear writing and exposition:** The paper is written very well and the concepts are described with clarity.
* **Formalization is done well:** The way the theoretical results are phrased and derived look sound. The proofs also appear to be correct.
* **Bounds regarding AFC is novel:** The bounds regarding AFC (lemma 2.9 and Theorem 2.10) appear to be novel. The concept of AFC seem to be useful when dealing with certain types of datasets where the "a set of features is strongly concentrated in a few data points in one class and spread out over almost all data points in another." Theorem 2.10, as far as I know, is interesting and novel.


****

**WEAKNESSES:**
* **Missing crucial references:** The submission doesn't cite related work that, in aggregate, cover a big portion of the claimed contributions.
  * "A game theoretic approach to class-wise selective rationalization.": [1] This paper proposes an interpretability framework that describes an extremely similar decision pipeline (adversarial and collaborative agents and a "verifier"). **This paper also provides bounds mutual information on the features and the labels** that seem to be analogous to the results reported in the current submission (though not identical0. Both submissions constrain the provers as feature selectors (i.e. masks on all input features) and work with very similar loss functions. There are subtle differences (i.e. the submission allows the verifier to output "I don't know " explicitly), but these don't represent conceptual disagreements. The "cheating" failure mode described in the submission is described as "degeneration" in this submission.
  * "Learning to Give Checkable Answers with Prover-Verifier Games.": [2] This submission **also explicitly proposes a framework with a prover and a verifier aimed at discovering complete and sound proof-verification protocols**. Here, the collaborative and adversarial components are condensed in a single prover with a different loss function. The prover isn't constrained to output masks in this case. This paper also explicitly links it's contributions to Interactive proof systems.
* **Limited novelty:** Given the above prior works (along with the "AI Safety via Debate" paper, which the authors cite), the conceptual contribution of this submission is significantly diminished. Contrary to what the authors claim, this is not the first method that achieves feature based interpretability.
* **Experiments are not real-world as claimed:** While the authors claim that they use real-world datasets to display the proposed algorithm and the theoretical bounds, the results are only limited to the UCI Census Income dataset (which is simple enough to support exhaustive search over selected features) for displaying the algorithm, and the MNIST dataset to display the theoretical bounds.
* **Proposed training algorithm is not practical:** Algorithm 1 (Merlin-Arthur Training) requires finding the argmin/argmax of the agents at each iteration, and appear to be very difficult to scale to more difficult/larger problems.
* **Theorems assume minimax optimality:** The proofs assume that an equilibrium of the proposed game is found. To complement this, an algorithm to check if an equilibrium has been found (or approximated to a sufficient degree) is needed. Solutions that appear to be near an equilibrium might behave significantly differently than how solutions at the equilibria could. This limitation, I believe, should be highlighted in the Limitations. (happy to be

****

**QUESTIONS TO AUTHORS:**
* **About resistance to existing interpretability manipulation techniques:** Among the contributions, you list "We show that our setup is resistant to existing interpretability manipulation techniques.". Do you mind elaborating on which experiments support this claim?
* **The "I don't know" option:** Is allowing Arthur to output "I don't know" essential in establishing the theoretical/empirical results? It seems like assigning uniform probability to all output logits would be an equivalent way for a classifier to say "I don't know". Is there a clear benefit to keeping having this as a separate option?
* **Question about Example 2.2** It seems like $y$ isn't defined on coordinates that are not in $S$. Later in the experiments, you mention that these coordinates can be an arbitrary value or 0s. Just wanted to confirm if the definition somehow subsumes this.
* **AFC of real world datasets:** Why do you think the AFC of real-world datasets will likely have and AFC of 1?
* **Why remove relationship and marital status features?** Could you explain why it was necessary to remove relationship and marital status features features from the dataset? In many cases removing such separate but highly correlated features is not possible (either because feature separation isn't possible in the input space, or because one doesn't know which features are highly correlated. Time permitting, could you show the results when you don't remove these features? How does the performed protocol behave in that setup?
* **Are the results on UCI and MNIST datasets evaluated on a separate test set?** Just wanted to check if the trained feature extractors are evaluated on an unseen split.



[1] Chang, Shiyu, et al. "A game theoretic approach to class-wise selective rationalization." Advances in neural information processing systems 32 (2019).
[2] Anil, Cem, et al. "Learning to Give Checkable Answers with Prover-Verifier Games." arXiv preprint arXiv:2108.12099 (2021).



**Summary Of The Paper:**

The authors describe a classification framework that consists of a verifier (Arthur) and two provers (one collaborative [Merlin] and one adversarial [Morgana]). The provers aim to convince/trick the verifier in making a correct/incorrect classification.

**Theoretical results:**
* **Mutual information bound at equilibria:** The authors prove that if Arthur has low error rate, then at the minimax equilibrium, the features selected by Merlin have very high mutual information with the label.
* **Bound on average precision in terms of dataset Asymmetric Feature Concentration (AFC), relative success rate and class imbalance:** The authors give a bound on average precision using a couple concepts they define, such as AFC and relative success rate.

**Empirical results:**
* **Training algorithm:** The authors provide an algorithm to solve for the equilibria of the proposed framework.
* **Results:** The soundness of the framework and the proposed bounds are tested on a low-dimensional tabular dataset (UCI dataset and MNIST).

**Contributions:**
* **Theoretical results:** Derives bounds on the mutual information between the label and the features selected by the framework described in the paper (bounds are quite similar to those derived by [1]).
* **Defining Asymmetric Feature Concentration:** The authors define "asymmetric feature concentration", a quantity that characterizes when  "a set of features is strongly concentrated in a few data points in one class and spread out over almost all data points in another". They later use this to bound the average precision of a given feature selector. Both this concept and this bound (as far as I now) are novel
* **Experiments:** Demonstrates the proposed method on a low-dimensional tabular dataset (UCI data), and the theoretical bounds on MNIST.

[1] Chang, Shiyu, et al. "A game theoretic approach to class-wise selective rationalization." Advances in neural information processing systems 32 (2019).

**Summary Of The Review:**

This paper presents a promising framework to address an important and timely problem. It is written well and some of the theoretical results are novel and interesting.

The main issues with the submission are lack of novelty (both regarding the framework and the theoretical results) and a sufficient discussion of existing literature, which form the basis of my recommendation. A (potentially significant) rewrite that places the proposed framework/results within the existing literature and puts the emphasis on the theoretical results (especially revolving around AFC) would, in my opinion, make this submission a worthwhile contribution to ICLR.

---

> ### Author Response · Authors · 2022-11-09
> **Repsonse to "Questions to authors"**
>
> We thank the reviewer for their suggestions and are happy to answer their questions.
>
> 1. In section 3.1. we have cited a series of papers that manipulate explanations by introducing an extra loss term for the network training. The interpretability methods can then be brought to highlight uninformative or misleading features. We show that this scheme is unsuccessful when an adversary is present, see Figure 5. The soundness requirement forces Merlin to either reveal the sensitive feature or drop the accuracy.
> 2. We could implement Arthur without this option and interpret being close to uniform as "Don't know". The theory would hold regardless. The primary purpose of having an "I don't know" option is ease of formulation and numerical implementation. Empirically, it is harder for the neural network to learn to balance the different outputs than to simply classify "Don't know!".
> 3. We think there is a misunderstanding. The $\mathbf{y}$ in Example 2.2 is a data point that is defined on all coordinates. What Merlin and Arthur exchange is a feature, defined as the set of all data points that share their values on the set $S$. We numerically realise this by masking the coordinates outside $S$ by any fixed value, e.g. 0. All data points that agree on $S$ then become numerically equal thus representing the set.
> 4. We make this assumption so that we can evaluate our bounds numerically. From Figure 4, it can be observed that when the AFC is large, each individual feature can indicate one class whereas a combination of the same features can indicate the opposite class. Our assumption of an AFC of 1, ensures that this issue does not occur, i.e., the class indicated by individual features is the same as that predicted by multiple. We will modify our assertion of this fact in the paper.
> 5.We removed 'marital status' and 'relationship' to make the effect of the cheating prevention clearer to the reader. If we include them, then Merlin can communicate 'sex' up to an accuracy of 71\% via these features. But this does not constitute cheating in our sense, since these features do indeed have high mutual information with 'sex'.
> For an application in discrimination prevention, these informative features then have to be further investigated. But this issue is shared by all feature-based interpretability approaches and goes far beyond the scope of this paper. Nevertheless, we are thankful for this comment and agree to include the results with these features in the appendix.
> 6. MNIST and the UCI dataset come with distinct train and test sets. We always used the former for training and tested on the latter.

---

> > ### Comment · Reviewer_Pt1p · 2022-11-15
> > **Thank you for your response**
> >
> > Thank you for your answers and clarifications. The answers and the proposed changes address some of my key concerns. Especially, 1) highlighting in the paper how the theoretical results connect to those of Chang et. al., (especially how the current submission doesn’t assume feature independence, which is an important advantage) 2) removing novelty claims regarding the framework and the guarantees, and a discussion of existing work highlighted in the review above, 3) adding results on the UCI dataset that does include relationship and marital status, 4) highlighting the fact that the Theorem doesn’t assume an equilibrium, but only the optimality of Morgana and Arthur given a Merlin, would significantly improve the paper.
> >
> > The current version of the paper submitted to Openreview don’t appear to contain these modifications (Is this right?). Would it be possible for the authors to upload a revision to the extent that's possible given the time left* so that the reviewers can base their decisions on a the newer version during reviewer discussion period?
> >
> > * perhaps excluding any pending results on the UCI dataset, which I trust the authors will add.

---

> > > ### Author Response · Authors · 2022-11-15
> > > **Further Response**
> > >
> > > Thank you the helpful remarks! We have uploaded a revision of our paper with all changes marked.
> > > This includes also the additional results on the UCI dataset in the appendix C4.

---

> > > > ### Comment · Reviewer_Pt1p · 2022-11-18
> > > > **Thanks again - one additional remark**
> > > >
> > > > Thank you for uploading the revised paper with all the modifications clearly highlighted. I have gone over the changes, and they address some of the important issues highlighted in my review.
> > > >
> > > > One additional comment regarding the related works section: The game that Chang et. al. propose has six players (or 3 * number-of-classes players) where each class gets it's own two provers (honest and counterfactual). In Anil et al.'s setup there are only two players, where the prover's loss is simultaneously collaborative and adversarial with the verifier. To the extent I understand the subtleties of these works and the one currently under submission, all of these setups are motivated by similar considerations (i.e. that the prover shouldn't be able to sent spurious features to the verifier) but the differences in the game setups might have practical consequences (i.e. ease of optimization etc. ) -- hence might be worthy of mention in the Related Works section.

---

> > > > > ### Author Response · Authors · 2022-11-18
> > > > > **Our Response**
> > > > >
> > > > > Thank you for the remark and the further interest!
> > > > >
> > > > > Indeed, on a theoretical level these setups are equivalent. On a numerical level there is an important difference. We have added a section A5 in the appendix to explain this difference and justify our approach. We link to this comparison in the beginning of the numerics section, since there it becomes relevant for the first time. We summarise the statement here:
> > > > >
> > > > > The base approach is the one taken by Chang et al where verifiers and provers operate wrt a specific class. To scale this up they propose 3 players for every class.
> > > > > We combine these players over all classes to have a single Merlin, Arthur and Morgana. Arthur rejecting all classes is captured in the "Don't know!" option.
> > > > > This has two advantages: First, it basically amounts to weight-sharing to have all verifiers in a single network. The same holds for the provers. This means more data per network weight. The second is we can use our additional knowledge that there is only a single class for each datapoint, and have Arthurs last layer output a probability distribution over all classes (+ "Don't know!"). Both are well established approaches for multi-class deep learning.
> > > > >
> > > > > Anil et al go even further and combine the two provers into a probabilitstic prover that outputs a certificate for a random class. While this allows for even further weight-sharing it also has a very practical disadvantage. The probability of outputting the certificate for the correct class is the inverse of the number of classes. So in application after training one will only occasionally get the right classification, most of the time the prover produces a certificate that does not convince the verifier. In our setting we can simply use Merlin together with Arthur and get the correct classification with the right certificate. We can reserve the use of Morgana for the cases where we are interested in a counter-factual.

---

### Official Review · Reviewer_YnKy · 2022-11-03

**Confidence:** 3
**Correctness:** 2
**Technical Novelty And Significance:** 2
**Empirical Novelty And Significance:** 2
**Recommendation:** 3

**Clarity, Quality, Novelty And Reproducibility:**

Clarity: can be improved significantly, see my comments above.
Quality and novelty: a similar idea was introduced in [1], and the authors need to address it properly.
Reproducibility: Though implementation details are largely missing in the main paper, the authors reported details in the appendix and also provided code.

**Strength And Weaknesses:**

Strengths:
- The paper studies an important problem with potential high impact on AI safety.
- The connections between informations metrics to completeness and soundness are quite interesting

Weaknesses:
- The authors missed one existing work of similar idea. The authors claim many times in the paper about "for the first time" in some of their contributions. However, the idea of prover-verifier game is not new and has been carefully discussed in [1]. In particular, [1] proposed a prover-verifier game to encourage learning agents to solve decision problems in a verifiable manner, which overlaps significantly with this paper.
- The paper is hard to follow, especially for the technical parts. The authors should consider improving the writing. Particularly, I find the notations in section 2 a bit awkward. For example, the notation of $x \in \phi$ was used to denote a data point $x$ contains a feature $\phi$, which is quite weird given we typically use to specify an element $x$ is in the set $\phi$. In addition, the authors use this notation in conditioning, which makes it even more confusing. Another example is the use of $D$ and $\mathcal{D}$ in Theorem 2.8.


Other comments and questions:
- What is the main usage of of average precision? The authors derived the bound for average precision, but why such a quantity is useful in interpretability?
- Should the loss of Morgana to be $$-\log \left(A(s \cdot x)_0 - A(s \cdot x)_{-c(x)} \right)$$?
- In Figure 5 (1), why only report the predictions of Merlin but not Morgana's?
- For Figure 6, it seems quite clear that the precision would go to 1 if one increase $k$. Does that suggest that it is trivial to get a tight bound in that regime?
- The game involves both cooperative and adversarial components. Often the case, such a game is difficult to solve (in terms of finding equilibrium)? Is this the case in your experiments?



Reference:
[1] Learning to Give Checkable Answers with Prover-Verifier Games, 2021.

**Summary Of The Paper:**

This paper proposed a multi-agent interactive game that provides formal interpretability for classification problem. In particular, the interactive game is inspired by interactive proof systems. In addition, the authors connected information measures such as conditional entropy and average precision to completeness and soundness metrics. On top of that, the authors introduced numerical implementation that works in a few settings.


**Summary Of The Review:**

The paper is interesting, but a similar idea has been discussed which I hope the authors could address in the next revision. With proper discussions and comparisons with the existing work and improved writing, I may consider increasing my score. Overall, the paper in its current form is not enough to warrant acceptance.

---

> ### Author Response · Authors · 2022-11-09
> **Response to "Other comments and questions"**
>
> We thank the reviewer for their time and detailed commentary and are happy to answer their questions.
>
> 1. The mutual information between class and feature can be bounded by the binary entropy of the average precision, see Appendix B1. Since the slope of the binary entropy function becomes infinite close to 0 and 1, this bound is cumbersome to state directly with respect to the mutual information. But average precision close to 1 implies mutual information close to the maximum (given by class entropy). Note that precision has been introduced as a feature-quality measure for interpretability before, see Ribeiro et al. in our references.
> 2. No. If this would be the case, then Arthur giving the correct prediction with probability 1 would have infinite loss. In our formulation, Arthur minimises a sum of the Merlin and the Morgana loss, whereas Morgana tries to maximise the Morgana loss.
> 3. We kept Morgana's strategy out to focus on the behaviour of Merlin and Arthur, while Morgana is ignored. However, you are right, Morgana's strategy is interesting in this case. In the cheating case (1b), she selects exactly the same features as Merlin, just reversed, to fool Arthur. This corresponds to our illustration in Figure 3a). We will add these bars to the graphic.
> 4. Large $k$ allows Merlin to reveal a significant part of the image to Arthur. This allows them to  exchange only very convincing certificates and cooperate with almost perfect completeness and soundness, i.e. $\epsilon_s, \epsilon_c \approx 0$. The bound then quickly goes to 1, which is the trivial upper limit of perfect precision. For smaller $k$, there likely exist no strategy that ensures perfect soundness and completeness. Since the bound is very general, it is pessimistic for this specific implementation.
> 5. Regarding our implementation: As we commented in the end of section 3, when $k$ is small and no perfect strategy exists, our implementation tends to oscillate between high soundness or high completeness. We will try to stabilise this in further research. As soon as sufficiently large image cutouts are allowed the setup converges after about 10 epochs of training.
> Regarding the theory: We would like to note that our Theorem 2.10. does not depend on any equilibria, only on the measured soundness and completeness. It thus applies as well to a snapshot during training.

---

### Author Response · Authors · 2022-11-09
**General Answer**

We are grateful to the reviewers for their detailed comments and especially for highlighting the papers by Chang et al. and Anil et al. We thus redact the statement that we are the first to prove guarantees for network-based classifiers from our paper.
Nevertheless, we believe that our work differs significantly from the aforementioned publications, as we will explain in detail, and contributes crucial new concepts and results to the field of XAI. In addition, we also address the concerns that have been raised about the AFC and the notation.

### Chang et al.

This work considers a prover-verifier setting similar to ours and shows that the optimal strategy for the prover is to select the features with the highest mutual information.
We will include this work in our discussion and confirm that it predates ours in connecting a prover-verifier setup and mutual information.

**Differences:**

1. Chang et al. show that the best strategy for the prover is to select the features with the highest mutual information. However, there is no guarantee that an implemented prover actually uses the optimal strategy. We do not assume this and instead measure completeness and soundness to draw a conclusion on feature quality.
2. In their analysis, the rationale selection scheme selects each word independently of the other words. This is a strong restriction, which we consider impractical. In text data words like `not' completely change the meaning of the following word. Similar issues arise with image data. We do not impose this restriction.
3. We do not rely on the assumption that the features are independently distributed which is unrealistic for text and image data. For our numerical evaluation of Theorem 2.10, we rely on the assumption that the AFC constant is small. We discuss this in more detail below.

### Anil et al.

Anil et al. also consider a prover-verifier setting similar to ours. An equilibrium between the players corresponds to perfect completeness and soundness. We will highlight this contribution in our related work section.

**Differences:**

1. This work draws a connection between **game equilibria** and **protocol quality** (completeness and soundness of the prover-verifier pair). We instead draw a connection between **protocol quality** and **feature quality**. Thus, this paper complements our work as part of a more general argument.
2. Crucially, our work does not rely on equilibria. Theorem 2.8 holds for any strategy for Merlin, even with better ones available. For Theorem 2.10, neither player is required to play optimally. Since we measure completeness and soundness on the test dataset we do not need to make assumptions about being in equilibrium.
The work of Anil et al. rather justifies why we should expect to observe good completeness and soundness after training.
3. The assumption in Anil et al. that a perfect protocol is available to choose from is very strong. Our setup also works when this is not the case. Especially in Figure 4, for small feature size $k$, we observe the regime where there likely is no strategy with perfect completeness and soundness. Our bounds hold regardless.

### Asymmetric Feature Correlation (AFC)

Intuitively, a small AFC means: When feature 1 and feature 2 individually convince you of class A, they should not collectively convince you of class B. See our Figure 4 for an example of large AFC for the fish and fruit features.
Assuming feature independence (i.e. no correlation at all), as in Chang et al. also implies a small AFC, since the features cannot be differently correlated for different classes.

We argue that our bound, in terms of the AFC, is preferable to the assumption of no correlation, for the following reasons:

1. Almost all data, especially image and text, have strong correlations between features.
2. We argue that no correlation is a strictly stronger assumption than small AFC.
3. Our bound with the AFC is quantitative, i.e., it also applies when the AFC is larger than 1. An equivalent version for Chang et al. would need some measure of feature correlation in the bound.

### Notation

We agree with the reviewers that our notation $\mathbf{x} \in \phi$, when $\mathbf{x}$ has feature $\phi$ needs time to get used to. We still think it is worth the effort.
Our definition has two advantages:

1. The setup is easily extendable to features that are not simply maskings of images or text. Many references by the reviewers such as Anil et al. or ref. [5] of reviewer 3 treat more general queries. Many practitioners would define translations of maskings as essentially the same feature.
2. It allows us to reason about data and features as graphs, similar to ref. [3] of reviewer 3. We already made use of this in Appendix A.4, where we construct a concrete counter-example that shows that the debate setup of Irving et al. cannot lead to bounds of the same generality as ours.

---

> ### Author Response · Authors · 2022-11-15
> **Updates in the Revision**
>
> We have indicated all changes in blue in the document.
> List of changes made:
>
> 1. We redacted the claims that we are the first to provide theoretical guarantees for feature-based interpretability. The abstract, the contribution section and the conclusion now reflect that.
> 1. We added the references mentioned by the reviewers as well as further work related to them. We considerably extended the related work section to reflect the history of the interactive classification setup and highlight how our work improves upon existing publications.
> 1. We moved the definitions of the Asymmetric Feature Correlation and the relative success rate to the main paper. We also state more clearly that our assumption that these quantities are 1 are necessary for our numerical investigation.
> 1. We include Morgana's features in Figure 5 (1). We also added the same experiment including "relationship" and "marital status" in Appendix C4. We added the point that informative features still need to be investigated by social scientists to the ``Impact and Limitation'' section.
> 1. We also made the smaller changes suggested by the reviewers such as correcting typos, specifying the set $D$ explicitly, slimming the definition of the feature space etc.
>
> We thank the reviewers again for their constructive responses as they have significantly improved the paper.

---

### Decision · Program_Chairs · 2023-01-20

**Decision:**

Reject

**Justification For Why Not Higher Score:**

See above.

**Justification For Why Not Lower Score:**

N/A

**Metareview: Summary, Strengths And Weaknesses:**

The contribution here is a kind of prover-verifier game motivated by interpretability. A prover (sometimes honest and sometimes adversarial) selects a subset of features to show the verifier, and the verifier tries to guess the correct class based on these features. The approach is a variant of the prover-verifier games of Anil et al. (2021), the main difference being that there are separate provers without weight tying. Theoretical results show that the method finds features with high mutual information with the label.

Reviewers consider the general topic to be interesting and understudied. The M-A framing is appealing, although perhaps not very accurate as the M-A protocol centers on the verifier being able to exploit randomness, which isn't a feature of this setup.

The reviewers' main criticism was the omission of several closely related pieces of prior work which, in aggregate, covered most of the claimed contributions of this paper. The authors have updated the paper with the missing references. It appears, however, that the discussion of related work is still rather cursory, considering the amount of overlap. Direct experimental comparisons against prior work would be useful. It would also be worth clarifying which of the theoretical results depend on the novel setup; for instance, is there really a fundamental difference between the one-prover setup of Anil et al. (2021) and the current multi-prover setup that, e.g., makes a difference for the mutual information results?

I encourage the authors to account for the reviewers' feedback in a future submission.